# Zero-Shot Anomaly Detection via Batch Normalization

**Aodong Li**[*]
UC Irvine

**Chen Qiu**[*]
Bosch Center for AI

**Marius Kloft**
TU Kaiserslautern

**Padhraic Smyth**
UC Irvine

**Maja Rudolph**[†]
Bosch Center for AI

**Stephan Mandt**[†]
UC Irvine

## Abstract

Anomaly detection (AD) plays a crucial role in many safety-critical application domains. The challenge of adapting an anomaly detector to drift in the normal data distribution, especially when no training data is available for the "new normal", has led to the development of zero-shot AD techniques. In this paper, we propose a simple yet effective method called Adaptive Centered Representations (ACR) for zero-shot batch-level AD. Our approach trains off-the-shelf deep anomaly detectors (such as deep SVDD) to adapt to a set of inter-related training data distributions in combination with batch normalization, enabling automatic zero-shot generalization for unseen AD tasks. This simple recipe, batch normalization plus meta-training, is a highly effective and versatile tool. Our theoretical results guarantee the zero-shot generalization for unseen AD tasks; our empirical results demonstrate the first zero-shot AD results for tabular data and outperform existing methods in zero-shot anomaly detection and segmentation on image data from specialized domains. Code is at https://github.com/aodongli/zero-shot-ad-via-batch-norm

## 1 Introduction

Anomaly detection (AD)—the task of identifying data instances deviating from the norm [65]—plays a significant role in numerous application domains, such as fake review identification, bot detection in social networks, tumor recognition, and industrial fault detection. AD is particularly crucial in safety-critical applications where failing to recognize anomalies, for example, in a chemical plant or a self-driving car, can risk lives.

Consider a medical setting where an anomaly detector encounters a batch of medical images from different patients. The medical images have been recorded with a new imaging technology different from the training data, or the patients are from a demographic the anomaly detector has not been trained on. Our goal is to develop an anomaly detector that can still process such data using batches, assigning low scores to normal images and high scores to anomalies (i.e., images that differ systematically) without retraining. To achieve this zero-shot adaptation, we exploit the fact that anomalies are rare. Given a new batch of test data, a zero-shot AD method [18, 36, 51, 71] has to detect which features are typical of the majority of normal samples and which features are atypical.

We propose Adaptive Centered Representations (ACR), a lightweight zero-shot AD method that combines two simple ideas: batch normalization and meta-training. Assuming an overall majority of "normal" samples, a randomly-sampled batch will typically have more normal samples than anomalies. The effect of batch normalization is then to draw these normal samples closer to the center (in its recentering and scaling operation), while anomalies will end up further away from the center. Notably, this scaling and centering is robust to a distribution shift in the input, allowing a

---

[*]Equal contribution [†]Joint supervision.    Correspondence to: aodongl1@uci.edu

37th Conference on Neural Information Processing Systems (NeurIPS 2023).

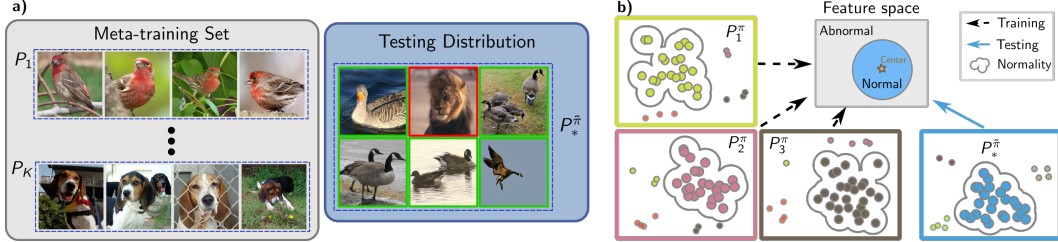

Figure 1: **a)** Demonstrations of concrete examples of a meta-training set and a testing distribution. It is not necessary for the meta-training set to include the exact types of samples encountered during testing. For instance, when detecting lions within geese, the training data does not need to include lions or geese. **b)** Illustration of zero-shot batch-level AD with ACR using a one-class classifier [63]. The approach encounters three tasks ($P_{1:3}^{\pi}$, Eq. (6)) during training (black arrows) and learns to map each task's majority of samples (i.e., the normal samples) to a shared learned center in embedding space. At test time (blue arrow), the learned model maps the normal (majority) samples to the same center and the distance from the center serves as AD score.

self-supervised anomaly detector to generalize to distributions never encountered during training. We propose a meta-training scheme to unlock the power of batch normalization layers for zero-shot AD. During training, the anomaly detector will see many different anomaly detection tasks, mixed from different choices for normal and abnormal examples. Through this variability in the training tasks, the anomaly detector will learn to rely as much as possible on the batch normalization operations in its architecture.

Advantages of ACR include that it is theoretically grounded, simple, domain-independent, and compatible with various backbone models commonly used in deep AD [56, 63]. Contrary to recent approaches based on foundation models [36], applicable only to images, ACR can be employed on data from any domain, such as time series, tabular data, or graphs.

We begin by presenting our assumptions and method in Sec. 2. Next, with the main idea in mind, we describe the related work in Sec. 3. We demonstrate the effectiveness of our method with experiments in Sec. 4. Finally, we conclude our work and state the limitations and societal impacts.

Our contributions can be summarized as follows:

- **An effective new method.** Our results for the first time show that training off-the-shelf deep anomaly detectors on a meta-training set, using batch normalization layers, gives automatic zero-shot generalization for AD. For which we derive a generalization bound on anomaly scores.
- **Zero-shot AD on tabular data.** We provide the first empirical study of zero-shot AD on tabular data, where our adaptation approach retains high accuracy.
- **Competitive results for images.** Our results demonstrate not only a substantial improvement in zero-shot AD performance for non-natural images, including medical imaging but also establish a new state-of-the-art in anomaly segmentation on the MVTec AD benchmark [5].

## 2 Method

We begin with the problem statement in Sec. 2.1 and then state the assumptions in Sec. 2.2. Finally we present our proposed solution in Sec. 2.3. The training procedure is outlined in Alg. 1 in Supp. C.

### 2.1 Problem Statement and Method Overview

We consider the problem of learning an anomaly detector that is required to immediately adapt (without any further training) when deployed in a new environment. The main idea is to use batch normalization as a mechanism for *adaptive batch-level* AD. For any batch of data containing mostly "normal" samples, each batch normalization shifts its inputs to the origin, thereby (1) enabling the discrimination between normal data and outliers/anomalies, and (2) bringing data from different distributions into a common frame of reference. (Notably, we propose applying batch norm in multiple layers for different anomaly scorers.) For the algorithm to generalize to unseen distributions,

we train our model on multiple data sets of "normal" data simultaneously, making sure each training batch contains a majority of related data points (from the same distribution) at a time.

Fig. 1a illustrates this idea, where all distributions are exemplified based on the example of homogeneous groups of animals (only dogs, only robins, etc.) The goal is to detect a lion among geese, where neither geese nor lions have been encountered before. Fig. 1b illustrates the scheme based on the popular example of deep support vector data description (DSVDD) [63], where samples are mapped to a pre-specified point in an embedding space and scored based on their distance to this point. All training distributions are mapped to the same point, as enabled through batch normalization.

## 2.2 Notation and Assumptions

To formalize the notion of a meta-training set, we consider a distribution of interrelated data distributions (previously referred to as groups) as commonly studied in meta-learning and zero-shot learning [3, 21, 22, 32]. This inter-relatedness can be expressed by assuming that $K$ training distributions $P_1, \ldots, P_K$ and a test distribution $P_*$ are sampled from a meta-distribution $\mathcal{Q}$:

$$P_1, \cdots, P_K, P_* \overset{\text{i.i.d.}}{\sim} \mathcal{Q}. \tag{1}$$

We assume that the distributions in $\mathcal{Q}$ share some common structure, such that training a model on one distribution has the potential to aid in deploying the model on another distribution. For example, the data $\mathbf{x}$ could be radiology images from patients, and each $P_j$ or $P_*$ could be a distribution of images from a specific hospital. These distributions share similarities but differ systematically because of differences in radiology equipment, calibration, and patient demographics[2]. Each of the distributions $P \in \mathcal{Q}$ defines a different anomaly detection task. For each task, we have to obtain an anomaly scoring function $S_\theta$ that assigns low scores to normal samples $\mathbf{x} \sim P$ and high scores to anomalies.

We now consider a batch $\mathcal{B} \subset \mathcal{D}$ of size $B$, taken from an underlying data set $\mathcal{D} \sim P$ of size $N$. The batch can be characterized by indexing data points from $\mathcal{D}$:

$$\mathcal{B} \equiv (i_1, ..., i_B) \sim \text{Unif}(\{1, ..., N\}). \tag{2}$$

We denote the anomaly scores on a *batch* level by defining a vector-valued anomaly score

$$\mathbf{S}_\theta(\mathbf{x}_\mathcal{B}) = (S_\theta^{i_1}(\mathbf{x}_\mathcal{B}), \cdots, S_\theta^{i_B}(\mathbf{x}_\mathcal{B})), \tag{3}$$

indicating the anomaly score for every datum in a batch. By thresholding the anomaly scores $S_\theta^i(\mathbf{x}_\mathcal{B})$, we obtain binary predictions of whether data point $\mathbf{x}_i$ is anomalous in *the context of batch* $\mathbf{x}_\mathcal{B}$.

By conditioning on a batch of samples, our approach obtains distributional information beyond a single sample. For example, an image of a cat may be normal in the context of a batch of cat images, but it may be anomalous in the context of a batch of otherwise dog images. This is different from current deep anomaly detection schemes that evaluate anomaly scores without referring to a context.

Before presenting a learning scheme of how to combine batch-level information in conjunction with established anomaly detection approaches, we discuss the assumptions that our approach makes. (The empirical or theoretical justifications as well as possibilities of removing or mitigating the assumptions can be found in Supp. A.)

**A1** *Availability of a meta-training set.* As discussed above, we assume the availability of a set of interrelated distributions. The meta-set is used to learn a model that can adapt without re-training.

**A2** *Batch-level anomaly detection.* As mentioned above, we assume we perform batch-level predictions at test time, allowing us to detect anomalies based on reference data in the batch.

**A3** *Majority of normal data.* We assume that normal data points take the majority in every i.i.d. sampled test batch.

Due to the absence of anomaly labels (or text descriptions) at test-time, we cannot infer the correct anomaly labels without assumptions **A2** and **A3**. Together, they instruct us that given a batch of test examples, the majority of the samples in the batch constitute normal samples.

---

[2]The divergence between distributions can be much larger than shown in this example. See our experiments.

## 2.3 Adaptively Centered Representations

**Batch Normalization as Adaptation Modules.** An important component of our method is batch normalization, which shifts and re-scales any data batch $\mathbf{x}_{\mathcal{B}}$ to have a sample mean zero and variance one. Batch normalization also provides a naïve parameter-free zero-shot batch-level anomaly detector:

$$S^i_{\text{naïve}}(\mathbf{x}_{\mathcal{B}}) = \|(\mathbf{x}_i - \bar{\mu}_{\mathbf{x}_{\mathcal{B}}})/\bar{\sigma}_{\mathbf{x}_{\mathcal{B}}}\|^2_2, \tag{4}$$

where $\bar{\mu}$ and $\bar{\sigma}^2$ are the coordinate-wise sample mean and sample variance. $\bar{\mu}$ is dominated by the majority of the batch, which by assumption A3, is the normal data. If the $\mathbf{x}_i$ lie in an informative feature space, anomalies will have a higher-than-usual distance to the mean, making the approach a simple, *adaptive* AD method, illustrated in Fig. 2 in Supp. D.

While the example provides a proof of concept, in practice, the normal samples typically do not concentrate around their mean in the raw data space. Next, we integrate this idea into neural networks and develop an approach that learns adaptively centered representations for zero-shot AD.

**Deep Models with Batch Normalization Layers as Scalable Zero-shot Anomaly Detectors.** In deep neural networks, the adaptation ability is obtained for *free* with batch normalization layers [35]. Batch normalization has become a standard and necessary component to facilitate optimization convergence in training neural networks. In common neural network architectures [27, 33, 60], batch normalization layers are used after each non-linear transformation layer, making a zero-shot adaptation with respect to its input batch. The entire neural network, stacking up many non-linear transformation and normalization layers, has powerful potential in scalable zero-shot adaptation and learning adaptation-needed feature representations for complex data forms.

**Training Objective.** As discussed above, we can instantiate $S_\theta$ as a deep neural network with batch normalization layers and optimize the neural network weights $\theta$. We first provide our objective function and then the rationality. Our approach is compatible with a wide range of deep anomaly detection objectives; therefore we consider a generic loss function $L[\mathbf{S}_\theta(\mathbf{x}_{\mathcal{B}})]$ that is a function of the anomaly score. For example, in many cases, the loss function to be minimized is the anomaly score itself (averaged over the batch).

The availability of a meta-data set (**A1**) gives rise to the following minimization problem:

$$\theta^* = \arg\min_\theta \frac{1}{K} \sum_{j=1}^K \mathbb{E}_{\mathbf{x}_{\mathcal{B}} \sim P_j} L[\mathbf{S}_\theta(\mathbf{x}_{\mathcal{B}})]. \tag{5}$$

Typical choices for $L[\mathbf{S}_\theta(\mathbf{x}_{\mathcal{B}})]$ include DSVDD [64] and neural transformation learning (NTL) [56]. Details and modifications of this objective will follow.

Why does it work? Batch normalization helps re-calibrate the data batches of different distributions into similar forms: normal data will center around the origin. Such calibration happens from granular features (lower layers) to high-level features (higher layers), resulting in powerful feature learning and adaptation ability[3]. We visualize the calibration in Fig. 3 in Supp. I.2. Therefore, optimizing Eq. (5) are able to learn a (locally) optimal $\mathbf{S}_{\theta*}$ that is adaptive to all $K$ *different* training distributions. Such learned adaptation ability will be guaranteed to generalize to unseen related distributions $P_*$. See Sec. 2.4 below and Supp. B for more details.

**Meta Outlier Exposure.** While Eq. (5) can be a viable objective, we can significantly improve over it while avoiding trivial solutions[4]. The approach builds on treating samples from other distributions as anomalies during training. The idea is that the synthetic anomalies can be used to guide learning a tighter decision boundary around the normal data [30]. Drawing on the notation from Eq. 1, we thus simulate a mixture distribution by contaminating each $P_j$ by admixing a fraction $(1-\pi) \ll 1$ of data from other available training distributions. The resulting corrupted distribution $P_j^\pi$ is thereby

$$P_j^\pi := \pi P_j + (1-\pi)\bar{P}_j, \qquad \bar{P}_j := \frac{1}{K-1} \sum_{i \neq j} P_i \tag{6}$$

This notation captures the case where the training distribution is free of anomalies ($\pi = 1$).

---

[3]Without batch normalization, optimizing Eq. (5) can be meaningless for some objectives. See Supp. I.1.

[4]We explain how this objective avoids trivial solutions in Supp. G and show the benefits in Tab. 4 of Supp. I.1.

Next, we discuss constructing an additional loss for the admixed anomalies, whose identity is known at training time. As discussed in [30, 58], many anomaly scores $\mathbf{S}_\theta(\mathbf{x}_\mathcal{B})$ allow for easily constructing a score $\mathbf{A}_\theta(\mathbf{x}_\mathcal{B})$ that behaves inversely. That means, we expect $\mathbf{A}_\theta(\mathbf{x}_\mathcal{B})$ to be *large* when evaluated on normal samples, and small for anomalies. Importantly, both scores share the same parameters. In the context of DSVDD, we define $\mathbf{S}_\theta(\mathbf{x}_\mathcal{B}) = 1/\mathbf{A}_\theta(\mathbf{x}_\mathcal{B})$, but other definitions are possible for alternative losses [58, 63, 64]. Using the inverse score, we can construct a supervised AD loss on the meta training set as follows.

We define a binary indicator variable $y_j^i$, indicating whether data point $i$ is normal or anomalous in the context of distribution $P_j$ (i.e., $y_j^i = 0$ iff $\mathbf{x}_\mathcal{B}^i \in P_j$). We later refer to it as *anomaly label*. A natural choice for the loss in Eq. (5) is therefore

$$L[\mathbf{S}_\theta(\mathbf{x}_\mathcal{B})] = \frac{1}{B} \sum_{i \in \mathcal{B}} \{(1 - y^i)\mathbf{S}_\theta^i(\mathbf{x}_\mathcal{B}) + y^i \mathbf{A}_\theta^i(\mathbf{x}_\mathcal{B})\}. \tag{7}$$

The loss function resembles the outlier exposure loss [30], but as opposed to using synthetically generated samples (typically only available for images), we use samples from the complement $\bar{P}_j$ at training time to synthesize outliers. The training pseudo-code is in Alg. 1 of Supp. C.

In addition to DSVDD, we also study backbone models such as binary classifiers and NTL [56]. For NTL, we adopt the $\mathbf{S}_\theta$ and $\mathbf{A}_\theta$ used by Qiu et al. [58]. For binary classifiers, we set $\mathbf{S}_\theta(\mathbf{x}) = -\log\big(1 - \sigma(f_\theta(\mathbf{x}))\big)$ and $\mathbf{A}_\theta(\mathbf{x}) = -\log \sigma(f_\theta(\mathbf{x}))$.

**Batch-level Prediction.**  After training, we deploy the model in an unseen production environment to detect anomalies in a zero-shot adaptive fashion. Similar to the training set, the distribution will be a mixture of new normal samples $P_*$ and an admixture of anomalies from a distribution never encountered before. For the method to work, we still assume that the majority of samples be normal (Assumption A3). Anomaly scores are assigned based on batches, as during training. For prediction, the anomaly scores are thresholded at a user-specified value.

Time complexity for prediction depends on the network complexity and is constant $O(1)$ relative to batch size, because the predictions can be trivially parallelized via modern deep learning libraries.

## 2.4  Theoretical Results

Having described our method, we now establish a theoretical basis for ACR by deriving a bounded generalization error on an unseen test distribution $P_*$. We define the generalization error in terms of training and testing losses, i.e., we are interested in whether the expected loss generalizes from the meta-training distributions $P_1, \cdots, P_K$ to an unseen distribution $P_*$.

To prepare the notations, we split $S_\theta(\mathbf{x})$ into two parts: a feature extractor $\mathbf{z} = f_\theta(\mathbf{x})$ that spans from the input layer to the last batch norm layer and that performs batch normalization, and an anomaly score $S(\mathbf{z})$ that covers all the remaining layers. We use $P_j^z$ to denote the data distribution $P_j$ transformed by the feature extractor $f_\theta$. We assume that $P_j^z$ satisfies $\mathbb{E}_{\mathbf{z} \sim P_j^z}[\mathbf{z}] = 0$ and $\mathrm{Var}_{\mathbf{z} \sim P_j^z}[\mathbf{z}] = 1$ for $j = 1, \ldots, K, *$ because $f_\theta$ ends up with a batch norm layer.

**Theorem 2.1.** *Assume the mini-batches are large enough such that, for batches from each given distribution $P_j$, the mini-batch means and variances are approximately constant across batches. Furthermore, assume the loss function $L[S(\mathbf{z})]$ is bounded by $C$ for any $\mathbf{z}$. Let $\|\cdot\|_{TV}$ denote the total variation. Then, the generalization error is upper bounded by*

$$\left| \mathbb{E}_{\mathbf{x}_\mathcal{B} \sim P_*} \left[ \frac{1}{B} \sum_{i=1}^B L[S_\theta^i(\mathbf{x}_\mathcal{B})] \right] - \frac{1}{K} \sum_{j=1}^K \mathbb{E}_{\mathbf{x}_\mathcal{B} \sim P_j} \left[ \frac{1}{B} \sum_{i=1}^B L[S_\theta^i(\mathbf{x}_\mathcal{B})] \right] \right| \le C \left\| P_*^z - \frac{1}{K} \sum_{j=1}^K P_j^z \right\|_{TV}.$$

The proof is shown in Supp. B. Note that Thm. 2.1 still holds if $P_j$ or $P_*$ are contaminated distributions $P_j^\pi$ or $P_*^{\tilde{\pi}}$.

**Remark.**  Thm. 2.1 suggests that the generalization error of the expected loss function is bounded by the total variation distance between $P_*^z$ and $\frac{1}{K} \sum_{j=1}^K P_j^z$. While we leave a formal bound of the TV distance to future studies, the following intuition holds: since $f_\theta$ contains batch norm layers, the empirical distributions $\frac{1}{K} \sum_{j=1}^K P_j^z$ and $P_*^z$ will share the same (zero) mean and (unit) variance. If both distributions are dominated by their first two moments, we can expect the total variation distance to be small, providing an explanation for the approach's favorable generalization performance.

## 3 Related Work

**Deep AD.** Many recent advances in AD are built on deep learning methods [65] and early strategies used autoencoder [9, 55, 85] or density-based [13, 66] models. Another pioneering stream of research combined one-class classification [70] with deep learning [57, 63]. Many other approaches to deep AD are self-supervised, employing a self-supervised loss function to train the detector and score anomalies [4, 23, 31, 44, 56, 68, 72, 75].

All of these approaches assume that the data distribution will not change too much at test time. However, in many practical scenarios, there will be significant shifts in the abnormal distribution and even the normal distribution. For example, Dragoi et al. [17] observed that existing AD methods fail in detecting anomalies when distribution shifts occur in network intrusion detection. Another line of work in this context requires test-time modeling for the entire test set, e.g., COPOD [47], ECOD [48], and robust autoencoder [85], preventing real-time deployment.

**Few-shot AD.** Several recent works have studied adapting an anomaly detector to shifts by fine-tuning a few test samples. One stream of research applies model-agnostic meta learning (MAML) [21] to various deep AD models, including one-class classification [22], generative adversarial networks [52], autoencoder [78], graph deviation networks [16], and supervised classifiers [20, 84]. Some approaches extend prototypical networks to few-shot AD [8, 40]. Kozerawski and Turk [39] learn a linear SVM with a few samples on top of a frozen pre-trained feature extractor, while Sheynin et al. [73] learn a hierarchical generative model from a few normal samples for image AD. Wang et al. [77] learn an energy model for AD. The anomalies are scored by the error of reconstructing their embeddings from a set of normal features that are adapted with a few test samples. Huang et al. [32] learn a category-agnostic model with multiple training categories (a meta set). At test time, a few normal samples from a novel category are used to establish an anomaly detector in the feature space. Huang et al. [32] does not exploit the presence of a meta-set to learn a stronger anomaly detector through synthetic outlier exposure. While meta-training for object-level anomaly detection (e.g., [22]) is generally simpler (it is easy to find anomaly examples, i.e., other objects different from the normal one), meta-training for anomaly segmentation (e.g., [32]) poses a harder task since image defects may differ from object to object (e.g., defects in transistors may not easily generalize to subtle defects in wood textures). Our experiments found that using images from different distributions as example anomalies during training is helpful for anomaly segmentation on MVTec-AD (see Supp. I.5).

In contrast to all of the existing few-shot AD methods, we propose a zero-shot AD method and demonstrate that the learned AD model can adapt itself to new tasks without any support samples.

**Zero-shot AD.** Foundation models pre-trained on massive training samples have achieved remarkable results on zero-shot tasks on images [37, 59, 82, 83]. For example, contrastive language-image pre-training (CLIP) [59] is a pre-trained language-vision model learned by aligning images and their paired text descriptions. One can achieve zero-shot image classification with CLIP by searching for the best-aligned text description of the test images. Esmaeilpour et al. [18] extend CLIP with a learnable text description generator for out-of-distribution detection. Liznerski et al. [51] apply CLIP for zero-shot AD and score the anomalies by comparing the alignment of test images with the correct text description of normal samples. In terms of anomaly segmentation, Trans-MM [7] is an interpretation method for Transformer-based architectures. Trans-MM uses the attention map to generate pixel-level masks of input images, which can be applied to CLIP. MaskCLIP [86] directly exploits CLIP's Transformer layer potential in semantic segmentation to generate pixel-level predictions given class descriptions. MAEDAY [71] uses the reconstruction error of a pre-trained masked autoencoder [28] to generate anomaly segmentation masks. WinCLIP [36], again using CLIP, slides a window over an image and inspects each patch to detect local defects defined by text descriptions.

However, foundation models have two constraints that do not exist in ACR. First, foundation models are not available for all data types. Foundation models do not exist for example for tabular data, which occurs widely in practice, for example in applications such as network security and industrial fault detection. Also, existing adaptations of foundation models for AD (e.g., CLIP) may generalize poorly to specific domains that have not been covered in their massive training samples. For example, Liznerski et al. [51] observed that CLIP performs poorly on non-natural images, such as MNIST digits. In contrast, ACR does not rely on a powerful pre-trained foundation model, enabling zero-shot AD on various data types. Second, human involvement is required for foundation models. While previous

pre-trained CLIP-based zero-shot AD methods adapt to new tasks through informative prompts given by human experts, our method enriches the zero-shot AD toolbox with a new adaptation strategy without human involvement. Our approach allows the anomaly detector to infer the new task/distribution based on a mini-batch of samples.

**Connections to Other Areas.** Our problem setup and assumptions share similarities with other research areas but differences are also pronounced. Those areas include *test-time adaptation* [10, 49, 53, 67, 76], *unsupervised domain adaptation* [38], *zero-shot classification* [79], *meta-learning* [21], and *contextual AD* [25]. Supp. H details the connections, similarities, and differences.

## 4 Experiments

We evaluate the proposed method ACR on both image (detection/segmentation) and tabular data, where distribution shifts occur at test time. We compare ACR with established baselines based on deep AD, zero-shot AD, and few-shot AD methods. The experiments show that our method is suitable for different data types, applicable to diverse AD models, robust to various anomaly ratios, and significantly outperforms existing baselines. We report results on image and tabular data in Sec. 4.1 and Sec. 4.2, and ablation studies in Sec. 4.3. Results on more datasets are in Supps. I.3 to I.6.

### 4.1 Experiments on Images

Visual AD consists of two major tasks: (image-level) anomaly detection and (pixel-level) anomaly segmentation. The former aims to accurately detect images of abnormal objects, e.g., detecting non-dog images; the latter focuses on detecting pixel-level local defects in an image, e.g., marking board wormholes. We test our method on both tasks and compare it to existing SOTA methods.

#### 4.1.1 Anomaly Detection

We evaluate ACR on images when applied to two simple backbone models: DSVDD [63] and a binary classifier. Our method is trained from scratch. The evaluation demonstrates that ACR achieves superior AD results on corrupted natural images, medical images, and other non-natural images.

**Datasets.** We study four image datasets: CIFAR100-C [29], OrganA [81] (and MNIST [42], and Omniglot [41] in Supp. I.4). We consider CIFAR100-C is the noise-corrupted version of CIFAR100's test data, thus considered as distributionally shifted data. We train using all training images from original CIFAR100 and test all models on CIFAR100-C. OrganA is a medical image dataset with 11 classes (for various body organs). We leave two successive classes out for testing and use the other classes for training. We repeat the evaluation on all combinations of two consecutive classes. Across all experiments, we apply the "one-vs-rest" setting at test time, i.e., one class is treated as normal, and all the other classes are abnormal [65]. We report the results averaged over all combinations.

**Baselines.** We compare our proposed method with a SOTA stationary deep anomaly detector (anomaly detection with an inductive bias (ADIB) [14]), a pre-trained classifier used for batch-level zero-shot AD (ResNet152 [27]), a SOTA zero-shot AD baseline (CLIP-AD [51]), and a few-shot AD baseline (one-class model-agnostic meta learning (OC-MAML) [22]). ResNet152-I and ResNet152-II differ in the which statistics they use in batch normalization: ResNet152-I uses the statistics from training and ResNet152-II uses the input batch's statistics. See Supp. E for more details.

**Implementation Details.** We set $\pi = 0.8$ in Eq. (6) to apply Meta Outlier Exposure. For each approach, we train a single model and test it on different anomaly ratios. Two backbone models are implemented: DSVDD [63] (ACR-DSVDD) and a binary classifier with cross entropy loss (ACR-BCE). More details are given in Supp. F.

**Results.** We report the results in terms of the AUROC averaged over five independent test runs with standard deviation. We apply the model to tasks with different anomaly ratios to study the robustness of ACR to the anomaly ratio at test time. Our method ACR significantly outperforms all baselines on Gaussian noise-corrupted CIFAR100-C and OrganA in Tab. 1. In Tabs. 8 and 9 in Supp. I, we systematically evaluate all methods on all 19 corrupted versions of CIFAR100 and on non-nature images (MNIST, Omniglot). The results show that on *non-natural* images (OrganA, MNIST, Omniglot) ACR performs the best among all compared methods, including the large pre-trained CLIP-AD baseline; on corrupted *natural* images (CIFAR-100C), ACR achieves results competitive

Table 1: AUC (%) with standard deviation for anomaly detection on CIFAR100-C with Gaussian noise [29] and medical image dataset, OrganA. ACR with both backbone models perform best.

| | CIFAR100-C | | | | OrganA | | |
|---|---|---|---|---|---|---|---|
| | 1% | 5% | 10% | 20% | 1% | 5% | 10% |
| ADIB [14] | 50.9±2.4 | 50.5±0.9 | 50.6±0.9 | 50.2±0.5 | 49.9±6.3 | 50.3±2.4 | 50.2±1.3 |
| ResNet152-I [27] | 75.6±2.3 | 73.2±1.3 | 73.2±0.8 | 69.9±0.6 | 54.2±1.1 | 53.9±0.5 | 53.2±0.6 |
| ResNet152-II [27] | 62.5±3.1 | 61.8±1.7 | 61.2±0.6 | 60.2±0.4 | 54.2±1.7 | 53.5±0.8 | 52.9±0.3 |
| OC-MAML [22] | 53.0±3.6 | 54.1±1.9 | 55.8±0.6 | 57.1±1.0 | 73.7±4.7 | 72.2±2.6 | 74.2±2.4 |
| CLIP-AD [51] | 82.3±1.1 | 82.6±0.9 | 82.3±0.9 | 82.6±0.1 | 52.6±0.8 | 51.9±0.6 | 51.5±0.2 |
| ACR-DSVDD (ours) | **87.7±1.4** | **86.3±0.9** | **85.9±0.4** | **85.6±0.4** | 79.0±1.0 | 77.7±0.4 | 76.3±0.3 |
| ACR-BCE (ours) | 84.3±2.2 | **86.0±0.3** | **86.0±0.2** | **85.7±0.4** | **81.1±0.8** | **79.5±0.4** | **78.3±0.3** |

Table 2: Pixel-level and image-level AUC (%) on MVTec AD. On average, our method outperforms the strongest baseline WinCLIP by 7.4% AUC in pixel-level anomaly segmentation.

| | MAEDAY [71] | CLIP [59] | Trans-MM [7] | MaskCLIP [86] | WinCLIP [36] | ACR (ours) |
|---|---|---|---|---|---|---|
| pixel-level | 69.4 | - | 57.5±0.0 | 63.7±0.0 | 85.1±0.0 | **92.5±0.2** |
| image-level | 74.5 | 74.0±0.0 | - | - | **91.8±0.0** | 85.8±0.6 |

with CLIP-AD and significantly outperforms other baselines. ACR is also robust on various anomaly ratios: without any (hyper)parameter tuning, the results are consistent and don't vary over 3%. The deep AD baseline, ADIB, doesn't have adaptation ability and thus fails to perform the testing tasks, leading to random guess results. Pre-trained ResNet152 armed with batch normalization layers can adapt but with limited ability, which is in contrast with our method that directly learns to adapt. Few-shot OC-MAML suffers because it requires a large support set at test time to achieve adaptation effectively. CLIP-AD has a strong performance on corrupted natural images but struggles with non-natural images, presumably because it is trained on massive natural images from the internet.

### 4.1.2 Anomaly Segmentation

We benchmark our method ACR on the MVTec AD dataset [5] in a zero-shot setup. Experiments show that ACR achieves new state-of-the-art anomaly segmentation performance.

**Datasets.** MVTec AD comprises 15 classes of images for industrial inspection. The goal is to detect the local defects accurately. To implement our method for zero-shot anomaly segmentation tasks, we train on the training sets of all classes except the target one and test on the test set of the target class. For example, when segmenting wormholes on wood boards, we train a model on the other 14 classes' training data except for wood and later test on wood test set. This satisfies the zero-shot definition as the model doesn't see any wood data during training. We apply this procedure for all classes.

**Baselines.** We compare our method to four zero-shot anomaly segmentation baselines: Trans-MM [7], MaskCLIP [86], MAEDAY [71], and WinCLIP [36]. The details are described in Sec. 3. We report their results listed in Jeong et al. [36], Schwartz et al. [71].

**Implementation Details.** We first extract informative texture features using a sliding window, which corresponds to 2D convolutions. The convolution kernel is instantiated with the ones in a pre-trained ResNet. We follow the same data pre-processing steps of Cohen and Hoshen [12], Defard et al. [15], Rippel et al. [62] to extract the features (the third layer's output in our case) of WideResNet-50-2 pre-trained on ImageNet. Second, we detect anomalies in the extracted features in each window position with our ACR method. Specifically, each window position corresponds to one image patch. We stack into a batch the patches taken from a set of images that all share the same spatial position. For example, we may stack the top-left patch of all testing wood images into a batch and use ACR to detect anomalies in that batch. Finally, the window-wise anomaly scores are bilinearly interpolated to the original image size to get the pixel-level anomaly scores. In implementing meta outlier exposure, we tried two sources of outliers: one is noise-corrupted images, and the other is images of other classes. We report results of the former in the main paper and the latter in Supp. I.5. More implementation details are given in Supp. F.

**Results.** Similar to common practice, we report both the pixel-level and image-level results in Tab. 2. We use the largest pixel-level anomaly score as the image-level score. All methods are evaluated with

Table 3: AUC (%) with standard deviation for anomaly detection on Anoshift with different anomaly contamination rations (1% - 20%) and on different splitting strategies AVG and FAR [17]. ACR with either backbone model outperforms all baselines. Especially, under the distribution shift occuring in the FAR split, ACR is the only method that is significantly better than random guessing.

| | 1% | | 5% | | 10% | | 20% | |
|---|---|---|---|---|---|---|---|---|
| | FAR | AVG | FAR | AVG | FAR | AVG | FAR | AVG |
| OC-SVM [69] | 49.6±0.2 | 62.6±0.1 | 49.6±0.2 | 62.6±0.1 | 49.5±0.1 | 62.7±0.1 | 49.5±0.1 | 62.6±0.1 |
| IForest [50] | 25.8±0.4 | 54.6±0.2 | 26.1±0.1 | 54.7±0.1 | 26.0±0.1 | 54.6±0.1 | 26.0±0.1 | 54.7±0.1 |
| LOF [6] | 37.3±0.5 | 59.6±0.3 | 37.0±0.1 | 59.5±0.1 | 37.0±0.1 | 59.5±0.1 | 37.1±0.1 | 59.5±0.1 |
| KNN [61] | 45.0±0.3 | 70.8±0.1 | 45.3±0.2 | 70.9±0.1 | 45.1±0.1 | 70.8±0.1 | 45.2±0.1 | 70.8±0.1 |
| DSVDD [63] | 34.6±0.3 | 62.3±0.2 | 34.7±0.1 | 62.5±0.1 | 34.7±0.2 | 62.5±0.1 | 34.7±0.1 | 62.5±0.1 |
| AE [1] | 18.6±0.2 | 25.3±0.1 | 18.7±0.2 | 25.5±0.1 | 18.7±0.1 | 25.5±0.1 | 18.7±0.1 | 25.5±0.1 |
| LUNAR [24] | 24.5±0.4 | 38.3±0.4 | 24.6±0.1 | 38.6±0.2 | 24.7±0.1 | 38.7±0.1 | 24.6±0.1 | 38.6±0.1 |
| ICL [72] | 20.6±0.3 | 50.5±0.2 | 20.7±0.2 | 50.4±0.1 | 20.7±0.1 | 50.4±0.1 | 20.8±0.1 | 50.4±0.1 |
| NTL [56] | 40.7±0.3 | 57.0±0.1 | 40.9±0.2 | 57.1±0.1 | 41.0±0.1 | 57.1±0.1 | 41.0±0.1 | 57.1±0.1 |
| BERT-AD[17] | 28.6±0.3 | 64.6±0.2 | 28.7±0.1 | 64.6±0.1 | 28.7±0.1 | 64.6±0.1 | 28.7±0.1 | 64.7±0.1 |
| ACR-DSVDD (ours) | 62.0±0.5 | **74.0±0.2** | 61.3±0.1 | **73.3±0.1** | 60.4±0.1 | 72.5±0.1 | 59.1±0.1 | 71.2±0.1 |
| ACR-NTL (ours) | **62.5±0.2** | 73.4±0.1 | **62.2±0.1** | 73.2±0.1 | **62.3±0.1** | **73.1±0.1** | **62.0±0.1** | **72.7±0.1** |

the AUROC metric. It shows that 1) our method is competitive to the SOTA method in image-level detection tasks, and 2) it surpasses the best baseline WinCLIP by a large margin (7.4% AUC on average) in anomaly segmentation tasks, achieving a new SOTA performance and testifying the potential of our method. We report class-wise results in Supp. I.5.

## 4.2 Experiments on Tabular Data

Tabular data is an important data format in many real-world AD applications, e.g, network intrusion detection and malware detection. Distribution shifts in such data occur naturally over time (e.g., as new malware emerges) and grow over time. Existing zero-shot AD approaches [36, 51] are not applicable to tabular data. We evaluate ACR on tabular AD when applied to DSVDD and NTL. ACR achieves a new SOTA of zero-shot AD performance on tabular data with temporal distribution shifts.

**Datasets.** We evaluate all methods on two real-world tabular AD datasets Anoshift [17] and Malware [34] where data shifts over time. Anoshift is a data traffic dataset for network intrusion detection collected over ten years (2006-2015). We follow the preprocessing procedure and train/test split suggested in Dragoi et al. [17]. We train the model on normal data collected from 2006 to 2010 [5], and test on a mixture of normal and abnormal samples (with anomaly ratios varying from 1% to 20%) collected from 2011 to 2015. We also apply similar protocols on Malware [34], a dataset for detecting malicious computer programs, and provide details in Supp. I.6.

**Baselines.** We compare with state-of-the art deep and shallow detectors for tabular AD [2, 17, 26] and study their performance under test distribution shifts. The shallow AD baselines include OC-SVM [69], IForest [50], LOF [6], and KNN [61]. The deep AD baselines include DSVDD [63], Autoencoder (AE) [1], LUNAR [24], internal contrastive learning (ICL) [72], NTL [56], and BERT-AD [17]. We adopt the implementations from PyOD [26] or their official repositories.

**Implementation Details.** To formulate meta-training sets, we bin the data against their timestamps (year for Anoshift and month for Malware) so each bin corresponds to one training distribution $P_j$. The training tasks are mixed with normality ratio $\pi = 0.8$. To create more training tasks, we augment the data using attribute permutations, resulting in additional training distributions. These attribute permutations increase the variability of training tasks and encourage the model to learn permutation-invariant features. At test time, the attributes are not permuted. Details are in Supp. F.

**Results.** In Tab. 3, we report the results on Anoshift split into AVG (data from 2011 to 2015) and FAR (data from 2014 and 2015). The two splits show how the performance degrades from average (AVG) to when strong distribution shifts happen after a long time interval (FAR). The results of Malware with varying ratios are in Tab. 12 and Supp. I.6. We report average AUC with standard deviation over five independent test runs. The results on Anoshift and Malware show that ACR outperforms all baselines on all distribution-shifted settings. Remarkably, ACR is the only method

---

[5]validate on a mixture of normal and abnormal samples collected from 2006 to 2010

that clearly outperforms random guessing on shifted datasets (the FAR split in Anoshift and the test split in Malware). All baselines perform worse than random on shifted test sets even though they achieve strong results when there are no distribution shifts (see results in Alvarez et al. [2], Dragoi et al. [17], Han et al. [26]). This worse-than-random phenomenon is also verified in the benchmark paper AnoShift [17]. The reason is that in cyber-security applications (e.g., Anoshift and Malware), the attacks evolve adversarially. The anomalies (cyber attacks) are intentionally updated to be as similar to the normal data to spoof the firewalls. That's why static AD methods like KNN flip their predictions during test time and achieve worse than random performance. In terms of robustness, although ACR-DSVDD's performance degrades a little (within 3%) when the anomaly ratio increases, ACR-NTL is fairly robust to high anomaly ratios. The degradation is attributed to the fact that the majority of normal samples get blurred as the anomaly ratio increase, leading to noisy batch statistics.

### 4.3 Ablation Studies

We perform several ablation studies in Supp. I.1, including 1) demonstrating the benefit of the Meta Outlier Exposure loss, 2) studying the effect of batch normalization, and 3) analyzing the effects of the batch sizes and the number of meta-training classes. To show that Meta Outlier Exposure is a favorable option, we compare it against the one-class classification loss and a fine-tuned version of ResNet152 on domain-specific training data. Tab. 4 shows that our approach outperforms the two alternatives on two image datasets. To analyze the effect of batch normalization, we adjust batch normalization usage during training and testing listed in Tab. 5. More details and the studies on the batch size, the number of meta-training classes, other normalization techniques (LayerNorm, InstanceNorm, and GroupNorm), effects of batch norm position, and robustness of the mixing hyperparameter $\pi$ can be found in Supp. I.1.

## 5 Conclusion

We studied the problem of adapting a learned AD method to a new data distribution, where the concept of "normality" changed. Our method is a zero-shot approach and requires no training or fine-tuning to a new data set. We developed a new meta-training approach, where we trained an off-the-shelf deep AD method on a (meta-) set of interrelated datasets, adopting batch normalization in every layer, and used samples from the meta set as either normal samples and anomalies, depending on the context. We showed that the approach robustly generalized to new, unseen anomalies.

Our experiments on image and tabular data demonstrated superior zero-shot adaptation performance when no foundation model was available. We stress that this is an important result since many, if not most AD applications in the real world rely on specialized datasets: medical images, data from industrial assembly lines, malware data, network intrusion data etc. Existing foundation models often do not capture these data, as we showed. Ultimately, our analysis shows that relatively small modifications to model training (meta-learning, batch normalization, and providing artificial anomalies from the meta-set) will enable the deployment of existing models in zero-shot AD tasks.

**Limitations & Societal Impacts**   Our method depends on the three assumptions listed in Sec. 2. If those assumptions are broken, zero-shot adaptation cannot be assured.

Anomaly detectors are trained to detect atypical/under-represented data in a data set. Therefore, deploying an anomaly detector, e.g., in video surveillance, may ultimately discriminate against under-represented groups. Anomaly detection methods should therefore be critically reviewed when deployed on human data.

## Acknowledgements

SM acknowledges support by the National Science Foundation (NSF) under an NSF CAREER Award, award numbers 2003237 and 2007719, by the Department of Energy under grant DE-SC0022331, by the IARPA WRIVA program, and by gifts from Qualcomm and Disney. Part of this work was conducted within the DFG research unit FOR 5359 on Deep Learning on Sparse Chemical Process Data. PS was supported by the US National Science Foundation under awards 1900644 and 1927245 and by the National Institute of Health under awards R01-AG065330-02S1 and R01-LM013344. SM and PS were supported by the Hasso Plattner Institute (HPI) Research Center in Machine Learning

and Data Science at the University of California, Irvine. MK acknowledges support by the Carl-Zeiss Foundation, the DFG awards KL 2698/2-1, KL 2698/5-1, KL 2698/6-1, and KL 2698/7-1, and the BMBF awards 03|B0770E and 01|S21010C. We thank Eliot Wong-Toi for helpful feedback on the manuscript.

The Bosch Group is carbon neutral. Administration, manufacturing and research activities do no longer leave a carbon footprint. This also includes GPU clusters on which the experiments have been performed.

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

# A  Justifications of Assumptions A1-A3

As follows, we provide justifications for assumptions A1-A3. Following the justification, we also discuss possibilities to remove or mitigate the assumptions.

**A1**   Assuming an available meta-training set is widely adopted in few-shot learning or meta-learning [21, 22, 32, 54] and domain generalization [46, 80]. In practice, the meta-training set can be generated using available covariates. For example, for our tabular data experiment, we used the timestamps; in medical data, one could use data collected from different hospitals or different patients to obtain separate sets for meta-training; and in MVTec-AD, we used the other training classes except for the target class to form the training set. We also provided an ablation study on the number of classes in the meta-training set (Tab. 7). We found even in the extreme case where we only have one data class in the training set, the trained model still provides meaningful results.

There are multiple ways to mitigate this assumption. If one does not have a meta-training set at hand, one can train their model on a different but related dataset, e.g., train on Omniglot but test on MNIST (see results below under this setting. We still get decent AUC results on MNIST).

| Anomaly ratio | 1% | 5% | 10% | 20% |
|---|---|---|---|---|
| AUROC | 84.4±2.4 | 85.2±2.5 | 84.3±2.5 | 82.2±2.4 |

**A2**   Batch-level prediction is a common assumption used in robustness literature [10, 49, 53, 67, 76]. In addition, batch-level predictions are widely used in real life. For example, people examine Covid19 test samples at a batch level out of economic and time-efficiency considerations[6]. To relax this assumption, our method can easily be extended to score individual data by presetting the sample mean and variance in BatchNorm layers with a collection of data. These moments are then fixed when predicting new individual data. Empirically, to understand the impacts of the batch size on the prediction performance, we conducted an ablation study with as small a batch size as three in the experiments.

**A3**   Besides being supported by the intuition that anomalies are rare, this is consistent with most of the data used in the literature. ADBench[7] has 57 anomaly detection datasets (with an average anomaly ratio of 5%), all matching our assumption that the normal data take the majority in each dataset.

We provide a simple mathematical argument for the validity of **A3**, showing that a mini-batch with a majority of anomalies is very unlikely to be drawn for a sufficiently large mini-batch size $B$. Let $p < 1/2$ denote the fraction of anomalies among the data and define $\Delta = 1/2 - p > 0$. For every data point $\mathbf{x}_i$ in the batch, let $y_i \sim \text{Bernoulli}(p)$ encode whether $\mathbf{x}_i$ is normal ($y_i = 0$) or abnormal ($y_i = 1$). The variable $S_B := y_1 + \cdots + y_B$ thus counts the number of anomalies in the batch, so that $S_B < B/2$ means that the majority of the data is normal. We want to show that the violation of A3 is unlikely, that is, $P(S_B \geq B/2)$ is small. By Hoeffding's inequality, since $0 \leq y_i \leq 1$ for all $i$, it follows that $P(S_B \geq B/2) = P(S_B - \mathbb{E}[S_B] \geq B/2 - Bp) \leq \exp\left(-2B(0.5-p)^2\right) \leq \exp\left(-2B\Delta^2\right)$, which converges to zero exponentially fast when $B \to \infty$.

# B  Generalization to an Unseen Distribution $P_*$

This section aims to provide a proof for Thm. 2.1. Inspired by Fallah et al. [19], we derive an upper bound of the generalization error of our meta-training approach on unseen distributions. The error is described in terms of the data distributions transformed by batch-norm-involved feature extractors.

**Definition B.1.**   Given a sample space $\Omega$ and its $\sigma$-field $\mathcal{F}$, the total variation distance between two probability measure $P_i$ and $P_j$ defined on $\mathcal{F}$ is

$$\|P_i - P_j\|_{TV} = \sup_{A \in \mathcal{F}} |P_i(A) - P_j(A)| = \sup_{f:0 \leq f \leq 1} \left|\mathbb{E}_{x \sim P_i}[f(x)] - \mathbb{E}_{x \sim P_j}[f(x)]\right| \tag{8}$$

---

[6]https://www.fda.gov/medical-devices/coronavirus-covid-19-and-medical-devices/pooled-sample-testing-and-screening-testing-covid-19

[7]https://github.com/Minqi824/ADBench

Now we split the neural network into two parts: the first part is the layers before (including) the last batch normalization layer, referred to as feature extractor $\mathbf{z} = f_\theta(\mathbf{x})$, and the second part is the layers after the last batch normalization layer, namely the anomaly score map $S(\mathbf{z}) = S(f_\theta(\mathbf{x})) = S_\theta(\mathbf{x})$. $S(\mathbf{z})$ may involve learnable parameters, but we omit the notations for conciseness. The split allows us to separate the effects of batch normalization layers on the generalization error of unseen distributions.

Under the transformation of $f_\theta$ consisting of batch normalization layers, we have the data distribution transformed into the distribution of adaptively centered representations

$$P_j(\mathbf{x}) \stackrel{\mathbf{z}=f_\theta(\mathbf{x})}{\Longrightarrow} P_j^z(\mathbf{z}), \quad j = 1, \cdots, K, * \tag{9}$$

resulting in $P_j^z$ with $\mathbb{E}_{P_j^z}[\mathbf{z}] = 0$ and $\mathrm{Var}_{P_j^z}[\mathbf{z}] = 1$.

Assume the mini-batch is large enough so that the mini-batch means and variances are approximately constant across batches, i.e., the batch statistics in batch normalization layers are equal to the population-truth values. Consequently, when $\mathbf{x}_1, \ldots, \mathbf{x}_B \stackrel{\text{i.i.d.}}{\sim} P_j$ which constitutes $\mathbf{x}_\mathcal{B}$, their latent representations $\mathbf{z}_i := \mathbf{f}_\theta^i(\mathbf{x}_\mathcal{B}) \stackrel{\text{i.i.d.}}{\sim} P_j^z$. for $i = 1, \cdots, B$. Then the expectation of the batch-level losses are

$$\mathbb{E}_{\mathbf{x}_\mathcal{B} \sim P_j}\left[\frac{1}{B}\sum_{i=1}^{B} L[S_\theta^i(\mathbf{x}_\mathcal{B})]\right] = \mathbb{E}_{\{\mathbf{z}_i \sim P_j^z\}_{i=1}^B}\left[\frac{1}{B}\sum_{i=1}^{B} L[S(\mathbf{z}_i)]\right]$$

$$= \frac{1}{B}\sum_{i=1}^{B}\mathbb{E}_{\{\mathbf{z}_i \sim P_j^z\}_{i=1}^B}[L[S(\mathbf{z}_i)]]$$

$$= \mathbb{E}_{\mathbf{z} \sim P_j^z}[L[S(\mathbf{z})]] \tag{10}$$

**Assumption B.2.** For any parameters (if any), the loss function $L[S(\cdot)]$ is bounded by $C$.

We now quantify the generalization error to an unseen distribution $P_*$ by the difference between the expected loss of data batches of $P_*$ and the one of meta-training distribution.

$$\left|\mathbb{E}_{\mathbf{x}_\mathcal{B} \sim P_*}\left[\frac{1}{B}\sum_{i=1}^{B} L[S_\theta^i(\mathbf{x}_\mathcal{B})]\right] - \frac{1}{K}\sum_{j=1}^{K}\mathbb{E}_{\mathbf{x}_\mathcal{B} \sim P_j}\left[\frac{1}{B}\sum_{i=1}^{B} L[S_\theta^i(\mathbf{x}_\mathcal{B})]\right]\right| \tag{11}$$

$$= \left|\mathbb{E}_{\mathbf{z} \sim P_*^z}[L[S(\mathbf{z})]] - \frac{1}{K}\sum_{j=1}^{K}\mathbb{E}_{\mathbf{z} \sim P_j^z}[L[S(\mathbf{z})]]\right| \quad \text{(by Eq. (10))} \tag{12}$$

$$= C\left|\mathbb{E}_{\mathbf{z} \sim P_*^z}\left[\frac{L[S(\mathbf{z})]}{C}\right] - \frac{1}{K}\sum_{j=1}^{K}\mathbb{E}_{\mathbf{z} \sim P_j^z}\left[\frac{L[S(\mathbf{z})]}{C}\right]\right| \quad \text{(by Assumption B.2)} \tag{13}$$

$$\leq C\sup_{0 \leq L/C \leq 1}\left|\mathbb{E}_{\mathbf{z} \sim P_*^z}\left[\frac{L[S(\mathbf{z})]}{C}\right] - \frac{1}{K}\sum_{j=1}^{K}\mathbb{E}_{\mathbf{z} \sim P_j^z}\left[\frac{L[S(\mathbf{z})]}{C}\right]\right| \tag{14}$$

$$= C\left\|P_*^z - \frac{1}{K}\sum_{j=1}^{K} P_j^z\right\|_{TV} \quad \text{(by Definition B.1)} \tag{15}$$

This result suggests the generalization error of the loss function is bounded by the total variation distance between $P_*^z$ and $\frac{1}{K}\sum_{j=1}^{K} P_j^z$. The batch normalization re-calibrates *all* $P_j$ such that $P_j^z$ centers at the origin and has unit variance, making the distributions similar. Thus the total variation gets smaller after batch normalization, lowering the generalization error upper bound.

The limitation of this analysis is we assume the batch statistics are population-truth moments (mean and variance) in $\mathbf{z}_i \stackrel{\text{i.i.d.}}{\sim} P_j^z$. So we cannot analyze the effects of the batch size $B$ during training and testing. That said, we provide empirical evaluations on different batch sizes $B$ at test time in Supp. I.1.

## C Algorithm

The training procedure of our approach is simple and similar to any stochastic gradient-based optimization. The only modification is to take into account the existence of a meta-training set. See Alg. 1.

**Algorithm 1:** Training procedure of ACR

**Input** : $K$ interrelated training distributions $P_1, \cdots, P_K \overset{\text{i.i.d.}}{\sim} \mathcal{Q}$
           Mixting rate $\pi$
           Deep anomaly detector model parameters $\theta$
           Sub-sample size $M$
           Mini-batch size $|\mathcal{B}|$
           learning rate $\alpha$
           Number of training iterations $T$
**Output :** Optimized anomaly detector with parameter $\theta_T$
1 Randomly initialize $\theta$
2 Construct $P_1^\pi, \cdots, P_K^\pi$ (Eq. (6))
3 **for** *iteration $t$ in $[1, \cdots, T]$* **do**
4      Sample $M$ tasks $\{\mathbf{x}_{\mathcal{B}_m}\}_{m=1}^M$ from all $K$ task distributions $\{P_1^\pi, \cdots, P_K^\pi\}$
5      $\theta_t \leftarrow \theta_{t-1} - \alpha \nabla_\theta \frac{1}{M} \sum_{m=1}^M L(\mathbf{S}_{\theta_{t-1}}(\mathbf{x}_{\mathcal{B}_m}))$ (Eq. (7))
6 **end**

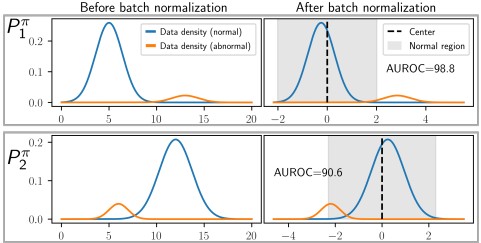

Figure 2: Illustration of batch normalization for AD with two tasks $P_1^\pi$ and $P_2^\pi$. The method (batch-)normalizes the data in $P_j^\pi$ separately. If each $P_j^\pi$ consists mainly of normal samples, most samples will be shifted close to the origin (by subtracting the respective task's mean). As a result, the samples from all tasks concentrate around the origin in a joint feature space (gray area) and thus can be tightly enclosed using, e.g., one-class classification. Samples from the test task are batch normalized in the same way.

## D   Toy Example with Batch Normalization

An important component of our method is batch normalization, which shifts and re-scales any data batch $\mathbf{x}_{\mathcal{B}}$ to have sample mean zero and variance one. Batch normalization also provides a basic parameter-free zero-shot batch-level anomaly detector (Eq. (4)). In Fig. 2, we show a 1D case of detecting anomalies in a mixture distribution. The mixture distribution composes of a normal data distribution (the major component) and an abnormal data distribution (the minor component). Eq. (4) adaptively detects anomalies at a batch level by shifting the normal data distribution toward the origin and pushing anomalies away. Setting a user-specified threshold allows making predictions.

## E   Baselines

**CLIP–AD [51].** CLIP (Contrastive Language–Image Pre-training [59]) is a pre-trained visual representation learning model that builds on open-source images and their natural language supervision signal. The resulting network projects visual images and language descriptions into the same feature space. The pre-trained model can provide meaningful representations for downstream tasks such as image classification and anomaly detection. When applying CLIP on zero-shot anomaly detection, CLIP prepares a pair of natural language descriptions for normal and abnormal data: $\{l_n$ = "A photo of {NORMAL_CLASS}", $l_a$ = "A photo of something"$\}$. The anomaly score of a test image $\boldsymbol{x}$ is the relative distance between $\boldsymbol{x}$ to $l_n$ and $\boldsymbol{x}$ to $l_a$ in the feature space,

$$s(\boldsymbol{x}) = \frac{\exp(\langle f_x(\boldsymbol{x}), f_l(l_a)\rangle)}{\sum_{c \in \{l_n, l_a\}} \exp(\langle f_x(\boldsymbol{x}), f_l(c)\rangle)},$$

where $f_x$ and $f_l$ are the CLIP image and description feature extractors and $\langle \cdot, \cdot \rangle$ is the inner product. We name this baseline CLIP-AD.

Compared to our proposed method, CLIP-AD requires a meaningful language description for the image. However, this is not always feasible for all image datasets like Omniglot [41], where people can't name the written characters easily. In addition, CLIP-AD doesn't apply to other data types like tabular data or time-series data. Finally, CLIP-AD has limited ability to adapt to a different data distribution other than its training one. These limitations are demonstrated in our experiments.

**OC-MAML [22].** One-Class Model Agnostic Meta Learning (OC-MAML) is a meta-learning algorithm that taylors MAML [21] toward few-shot anomaly detection setup. OC-MAML learns a global model parameterization $\theta$ that can quickly adapt to unseen tasks with a few new task data points $S$, called a support set. The new-task adaptation takes the global model parameters to a task-specific parameterization $\phi(\theta, S_t)$ that has a low loss $L(Q_t; \phi(\theta, S_t))$ on the new task $t$, represented by another dataset $Q_t$, called a query set. OC-MAML uses a one-class support set to update the model parameters $\theta$ with a few gradient steps to get $\phi$. To learn an easy-to-adapt global parameterization $\theta$, OC-MAML directly minimizes the target loss on lots of training tasks. Suppose there are $T$ tasks for training. The following loss function is minimized

$$l(\theta) = \frac{1}{T} \sum_{t=1}^{T} \mathbb{E}_{S_t \sim p_S^t, Q_t \sim p_Q^t} [L(Q_t; \phi(\theta, S_t))], \tag{16}$$

where $p_S^t$ is task $t$'s support set distribution and $p_Q^t$ is the query set distribution. During training, the support set contains $K$ normal data points where $K$ is usually small, termed $K$-shot OC-MAML. The query set contains an equal number of normal and abnormal data and provides optimization gradients for $\theta$. During test time, OC-MAML adapts the global parameter $\theta$ on the unseen task's support set $S^*$, resulting in a task-specific parameter $\phi(\theta, S^*)$. The newly adapted parameters are then used for downstream tasks.

OC-MAML is not a zero-shot anomaly detector and requires $K$ support data points to adapt compared to our method. Our method is simpler in training as it doesn't need to adapt to the support set with additional gradient updates, characterized in the function $\phi(\theta, S)$. OC-MAML is also different in batch normalization. Rather than the original batch normalization, OC-MAML first computes the batch moments using the support set and then normalizes both the support and query set with the same moments. However, the computed moments can be noisy when the support set size is small. In our experiments, we adopt a 1-shot OC-MAML for all image data.

**ResNet152 [27].** Because batch normalization is an effective tool for zero-shot anomaly detection (see Fig. 1b), we directly apply batch normalization on extracted features from a pre-trained model. We then compute the anomaly score as the Euclidean distance between a feature vector and the origin in the feature space. Our experiments use a ResNet152 model pre-trained on ImageNet as a feature extractor and extract its 2048-dimension penultimate layer output as the final feature vector. Upon computing the features of an input batch through batch normalization layers in ResNet, two variants are available: using the batch statistics from training or re-computing the statistics of the test input batch itself. We name the former variant ResNet152-I and the latter ResNet152-II. Baseline ResNet152 doesn't optimize the feature extractor jointly with the zero-shot detection property of batch normalization. Hence the extracted pre-trained features are not optimal for the zero-shot AD.

**ADIB [14].** In addition to zero-shot and few-shot anomaly detectors, we also compare with the state-of-the-art deep anomaly detector ADIB [14] which use pre-trained image features and additional data for outlier exposure in training. We use a "debiased" subset of TinyImageNet as the outlier exposure data for CIFAR100 as suggested in Hendrycks et al. [30], use EMNIST [11] as the outlier exposure data for MNIST as suggested in Liznerski et al. [51], use OrganC and OrganS datasets [81] as outlier exposure data for OrganA, and use half of the training data as normal data and half of the training data as auxiliary outliers for Omniglot.

## F Implementation Details

**Practical Training and Testing.** On visual anomaly classification and tabular AD, we construct training and test distributions using labeled datasets[8], where all $\mathbf{x}$ from the same class $j$ (e.g., all 0's in MNIST) are considered samples from the same $P_j$. The dataset $\mathcal{Q}$ (e.g., MNIST as a whole) is the meta-set of all these distributions.

For training and testing, we split the meta-dataset into disjoint subsets. In the MNIST example, we define $P_0, ..., P_4$ as the distributions of images with digits $0 - 4$ and use them for training. For testing, we select a single distribution of digits not seen during training (e.g., digit 5) as the "new normal" distribution $P_*$ to which we adapt the model. The remaining digits ($6 - 9$ in this example) are used as test-time anomalies. To reduce variance, we rotate the roles among digits $5 - 9$, using each digit as a test distribution once.[9]

### F.1 Implementation Details on Image Data for Anomaly Detection

**Hyperparameter Search.** We search the hyperparameters on a validation set split from the training set, after which we integrate the validation set into the training set and train the model on that. Then we test the model on the test set.

On CIFAR100-C, we construct the validation set on the training set of the primitive CIFAR100. We randomly select 20 classes as the validation set and set the remaining classes to be the training dataset at validation time. We search the neural network architecture (layers (3,4,5,6), number of convolutional kernels (32, 64, 128), and the output dimensions (8, 16, 32, 64, 128) while fixing the kernel size by 3x3. For the learning rate, we search values 0.1, 0.01, 0.001, 0.0001, and 0.00001, after which we search finer values in a binary search fashion. We also search the mini-batch size $B$ (30, 60) and the number of sub-sampled tasks $M$ (16, 32, 64) at each iteration. We select the combination that trade-off the convergence speed and optimization stability. When selecting the anomaly ratio $\pi$ (Eq. (6)), we test 0.99, 0.95, 0.9, 0.8, 0.6 and find the results are quite robust to these values. So we fix $\pi = 0.8$ across the experiments.

On non-natural image datasets (OrganA, Omniglot, and MNIST), we search hyperparameters on the validation set of Omniglot and use the searched hyperparameters on all datasets. Specifically, at validation time, we randomly split the Omniglot into 1200 classes for training and 423 classes for validation. After optimizing the hyperparameters, we constantly use the first 1200 classes for training and the remaining 423 classes for testing. The searched hyperparameters are the same as the ones in CIFAR100-C described above.

**Training Protocols.** We train the model 6,000 iterations on CIFAR100 data, 10,000 iterations on Omiglot, and 2,000 iterations on MNIST and OrganA. Each iteration contains 32 training tasks; each task mini-batch has 30 (for datasets other than CIFAR100) or 60 (for CIFAR100) points sampled from $P_j^{0.8}$. All 32 training tasks' gradients are averaged and incur one gradient update per iteration.

**ACR-DSVDD.** We use the standard convolutional neural network architecture used in meta-learning. Specifically, the network contains four convolution layers. Each convolution layer is followed by a batch normalization layer and a ReLU activation layer. The final layer is a fully-connectly layer followed by a batch normalization layer. The center of DSVDD has the same dimension as the output of the fully-connected layer, which is 32. For CIFAR100/CIFAR100-C, each convolution layer has 128 kernels. For MNIST, Omniglot, and OrganA, each convolution layer has 64 kernels. Each kernel's size is 3x3. We use Adam with a learning rate of $0.003$ on CIFAR100 dataset and $1e - 4$ on all the other datasets.

**ACR-BCE.** We use the same network structure as ACR-DSVDD without the final batch normalization layer and the center. The final fully-connected layer has output dimension of 1. We train the model with binary cross entropy loss. We use Adam with a learning rate of $0.003$ on CIFAR100 dataset and $1e - 4$ on all the other datasets.

---

[8]these are either classification datasets or datasets where one of the covariates is binned to provide classes.

[9]This is the popular "one-vs-rest" testing set-up, which is standard in AD benchmarking. (e.g., [65])

### F.2 Implementation Details on MVTec AD for Anomaly Segmentation

**Training Protocols.** Since the images are roughly aligned, we can use a sliding window over the image to detect local defects in each window. However, this requires pixel-level alignment and is unrealistic for the MVTec AD dataset. We instead first extract informative texture features using a sliding window, which corresponds to the 2D convolutions. The convolution kernel is instantiated with the ones in a pre-trained ResNet. We follow the same data pre-processing steps of Cohen and Hoshen [12], Defard et al. [15], Rippel et al. [62] to extract the texture representations (the third layer's output in our case) of WideResNet-50-2 pre-trained on ImageNet. Second, we detect anomalies in the extracted features in each sliding window position with our ACR method. Specifically, each window position corresponds to one image patch. We stack into a batch the patches taken from a set of images that all share the same spatial position in the image. For example, we may stack the top-left patch of all testing `wood` images into a batch and use ACR to detect anomalies in that batch. Finally, the window-wise anomaly scores are bilinearly interpolated to the size of the original image, i.e., the pixel-level anomaly scores.

Following the same data pre-processing steps of Cohen and Hoshen [12], Defard et al. [15], Rippel et al. [62], we extract the texture representations (the third layer's output in our case) of WideResNet-50-2 pre-trained on ImageNet. After feature extraction, a batch of $B$ images leads to a representation of size $(B, C, H, W) := (B, 1024, 14, 14)$. These representations contain both textual and spatial information. During meta-training, we treat each spatial position as one new class so that there are $14 \times 14 = 196$ new classes for each original class (e.g., `wood`), and each new class contains $B$ data points, each a 1024 long vector. As a result, the model (DSVDD in our usage) takes a batch of vectors of size $(B, 1024)$ as input and assigns anomaly scores to each vector within the batch. When adding synthetic abnormal data (Eq. (6)), we use Gaussian noise corrupted input vectors rather than new class data, incorporating the fact that local defects result in similar feature vectors instead of globally different textures. Specifically, we add Gaussian noise sampled from $\mathcal{N}(0, 0.01I)$ and set $\pi = 0.5$ in Eq. (6). Since there is no During testing, we batch each position $(h, w)$ of all images and detect anomalies at position $(h, w)$. Because the defects are local, and there is a possibility that there is no defect at some position $(h_0, w_0)$, we manually add synthetic noisy vectors into the tested batch as the training procedure to ensure the images have a low anomaly score at $(h_0, w_0)$. After getting anomaly scores, we remove the synthetic vector results, leading to scores of size $(B, 14, 14)$, and then upscale the scores into the original image size by bilinear interpolation. We acknowledge that using CutPaste [45] to generate more realistic synthetic abnormal samples is another option. We leave the investigation to future work.

**ACR-DSVDD and Hyperparameter Search.** Our model is a five-layer MLP with intermediate batch normalization layers and ReLU activations. The hidden sizes of the perceptrons are $[512, 256, 128, 64, 32]$. The center is size 32. The statistics of all batch normalization layers are computed on fly on the training/test batches. We average the gradients of 32 randomly sampled tasks for each parameter update. Each task contains 30 normal feature vectors and 30 noise-corrupted feature vectors. We train the model with Meta Outlier Exposure loss. We set the learning rate 0.0003 and iterate 50 updates for each class. We search the hyperparameters on a test subset of `bottle` class (half of the original test set) and apply the same hyperparameters to all classes afterward.

### F.3 Implementation Details on Tabular Data

ACR-NTL has the same model architecture as the baseline NTL, and ACR-DSVDD adds one additional batch normalization layer on top of the DSVDD baseline. Our algorithm is applicable to the existing backbone models without complex modifications.

**ACR-DSVDD.** ACR is applied to the backbone model DSVDD [63]. The neural network of DSVDD is a four-layer MLP with intermediate batch normalization layers and ReLU activations. The hidden sizes on Anoshift dataset are $[128, 128, 128, 32]$. The hidden sizes on Malware dataset are $[64, 64, 64, 32]$. One batch normalization layer is added on the top of the network on Anoshift experiment. The statistics of all batch normalization layers are computed on fly on the training/test batches. We use Adam with a learning rate of $4e-4$ on Anoshift dataset and $1e-4$ on Malware dataset.

**ACR-NTL.** ACR is applied to the backbone model NTL [56]. The shared encoder of NTL is a four-layer MLP with intermediate batch normalization layers and ReLU activations. The hidden sizes of the encoder are $[128, 128, 128, 32]$. The statistics of all batch normalization layers are computed on fly on the training/test batches. We set the number of neural transformations as $19$. Each neural transformation is parametrized by a three-layer MLP of hidden size of $128$ with ReLU activations. All networks are optimized jointly with Adam with a learning rate of $4e - 4$.

# G   Meta Outlier Exposure Avoids Trivial Solutions.

The benefit of the outlier exposure loss in meta-training is that the learning algorithm cannot simply learn a model on the *average* data distribution, i.e., without learning to adapt. This failure to adapt is a common problem in meta-learning. Our solution relies on using each training sample $\mathbf{x}_i$ in different contexts: depending on the sign of $y_{i,j}$, data point $\mathbf{x}_i$ is considered normal (when drawn from $P_j$) or anomalous (when drawn from $\bar{P}_j$). This ambiguity prevents the model from learning an average model over the meta data set and forces it to adapt to individual distributions instead.

For example, DSVDD with its original loss function may suffer from a trivial solution that maps any input data to the origin in the feature space and achieves the optimal zero loss [63]. This trivial solution is also possible in our proposed meta-training procedure. But Meta Outlier Exposure gets rid of this trivial solution because mapping everything to the origin incurs an infinite loss on $\mathbf{A}_\theta$. Similar reasoning also applies to binary cross entropy loss.

# H   Connections to Other Areas

Our problem setup and assumptions share similarities with other research areas but differences are also pronounced.

**Connection to Batch Normalization-Based Test-time Adaptation (TTA).** Many works for TTA feature batch-level predictions [10, 49, 53, 67, 76] assumes its test-time data are corrupted but from the *same semantic classes* as the training data, but the zero-shot AD's test data can be drawn from a completely new class.

**Connection to Unsupervised Domain Adaptation (UDA).** Although our approach uses unlabeled data like the UDA setting [38], UDA assumes the unlabeled data from the shifted domain is available during training and can be used to update the model parameters. But our method doesn't rely on the availability of novel data during training time and doesn't require updating the model parameters during test time.

**Connection to Zero-shot Classification.** Xian et al. [79] explains the nature of zero-shot classification and writes that "the crux of the matter for all zero-shot learning methods is to associate observed and non-observed classes through some form of auxiliary information which encodes visually distinguishing properties of objects." The *auxiliary information* demands extra human annotations like picture attributes. In contrast, our method assumes a batch of test data without human annotations. The test distribution information is automatically contained in the batch statistics.

**Connection to Meta-learning.** Although we assume that there is a meta-training dataset available like meta-learning, we don't require a support set for updating the model during both training time and test time. The presence of a support set differentiates our method from meta-learning in many aspects. First, for the most well-known technique (MAML) in meta-learning, training requires second-order derivative information of the support set loss function, which is computationally expensive and slows the optimization. Second, it is unclear how to select the support set size for adaptation. Sometimes, it may require a large support set to achieve good adaptations. For example, OC-MAML needs at least a 10-shot support set to perform on par with our method on CIFAR100-C; Third, the support set requires labeled data. This already adds burdens to practitioners. Fourth, the model parameter updates require additional maintenance and extra cost during testing.

**Connection to Contextual AD.** Contextual AD considers a changing notion of normality based on context [25, 74]. In contextual AD, the training and testing data are from the *same* data generating process, which involves (hidden or observed) contextual variables controlling the generation. This is different from our setup. We tackle the problem when the training and testing data are from different data generating processes.

Table 4: AUC (%) with standard deviation for anomaly detection on CIFAR100-C and Omniglot. As an ablation, rather than utilizing outlier exposure, we trained Zero-shot BN only on normal data of each task.

| | CIFAR100-C (Gaussian Noise) | | | | Omniglot | | |
|---|---|---|---|---|---|---|---|
| | 1% | 5% | 10% | 20% | 5% | 10% | 20% |
| One-class loss | 72.2±2.2 | 73.9±1.4 | 74.2±0.9 | 73.8±0.3 | 96.2±1.0 | 96.4±0.8 | 96.2±0.8 |
| (data-adapted) ResNet152 | 70.9±2.2 | 67.6±0.2 | 67.0±0.7 | 64.9±0.5 | 99.2±0.2 | 99.1±0.1 | 99.0±0.1 |

Table 5: The effects of batch normalization for zero-shot AD. The first two columns show different combinations of batch normalization usage during training and testing. The third column answers the question whether the type of batchnorm usage works for zero-shot AD.

| BatchNorm (train) | BatchNorm (test) | Work? |
|---|---|---|
| ✓ | ✓ | Yes |
| ✓ | ✗ | No |
| ✗ | ✓ | No |
| ✗ | ✗ | No |

Table 6: The effects of test time batch size on the results of zero-shot AD. We report the test results in AUC when the contamination ratio is set to 5% and 10%. The studies are conducted on the Gaussian noise version of CIFAR-100C. On the extreme batch sizes, each batch contains one anomaly.

| Batch size | 3 | 6 | 11 | 16 |
|---|---|---|---|---|
| One anomaly | 66.4±2.3 | 77.9±2.8 | 82.3±2.7 | 84.8±2.0 |

| Batch size | 20 | 40 | 60 | 80 | 100 |
|---|---|---|---|---|---|
| 5% | 83.7±1.9 | 85.3±1.1 | 85.6±1.2 | 85.9±0.8 | 85.6±0.7 |
| 10% | 84.5±1.5 | 86.1±0.8 | 85.7±0.7 | 85.8±0.5 | 85.8±0.6 |

Table 7: The effects of the number of classes used in training on zero-shot AD. We report the test results in AUC when the contamination ratio is fixed to 10%. The studies are conducted on the Omniglot dataset.

| #Training classes | 1 | 2 | 5 | 10 | 15 | | |
|---|---|---|---|---|---|---|---|
| AUROC | 59.0±0.6 | 71.8±0.6 | 72.5±0.3 | 72.2±1.0 | 75.3±0.4 | | |

| #Training classes | 20 | 40 | 80 | 160 | 320 | 640 | 1200 |
|---|---|---|---|---|---|---|---|
| AUROC | 79.0±1.0 | 90.5±0.5 | 95.3±0.2 | 97.6±0.2 | 98.1±0.2 | 98.4±0.1 | 99.1±0.2 |

# I   Additional Results

## I.1   Ablation Study

**Training with Different Losses.**   We study the benefits of using meta outlier exposure in Eq. (5) and compare to a) using one-class classification loss $L[\mathbf{S}_\theta(\mathbf{x}_{\mathcal{B}})] = \frac{1}{B} \sum_{i \in \mathcal{B}} \mathbf{S}_\theta^i(\mathbf{x}_{\mathcal{B}})$ with $\mathbf{S}_\theta^i(\mathbf{x}_{\mathcal{B}}) = \|\phi_\theta^i(\mathbf{x}_{\mathcal{B}}) - \mathbf{c}\|^2$ where $\phi_\theta$ is the feature map, b) (data-adapted) ResNet152. The data-adapted ResNet152 first learns the features by performing a multi-class classification task with the meta-training set. Then a batch normalization layer is applied on the top of penultimate layer representations for zero-shot anomaly detection. We train a 100-class classifier for CIFAR100C and Omniglot separately. Note that for Omniglot, we randomly sub-sample 100 classes from its 1400 training classes and train the classifier. From the results in Tab. 4 we can see that both ablations perform competitive with ACR on the simple Omniglot dataset, but perform much worse compared to ACR on the complex CIFAR100-C dataset. In conclusion, using meta outlier exposure in training is favorable.

**Training or Testing Without BatchNorm.**   We investigate whether training or testing without batch normalization works for zero-shot AD or not. To this end, we employ four different combinations

of batch normalization usage during training and testing and check which combination works and which doesn't. We trained the models with the same meta-training procedure as what we used in Sec. 4.1.1 and tested on CIFAR100-C and Omniglot. We present the results in Tab. 5. In the third column, "Yes" indicates the AUROC metric is significantly larger than 0.5, and therefore learns a meaningful zero-shot AD model; "No" indicates the AUROC performance is around 0.5, which means the predicted anomaly scores are just random guesses and the model cannot be used for zero-shot AD. Tab. 5 shows that only when the batch normalization is used both in training and testing, the zero-shot AD works. Otherwise, the meta-training procedure couldn't result in meaningful zero-shot AD representations.

Moreover, for the DSVDD model, we can theoretically show that training without batch normalization will not work with meta outlier exposure: the optimal loss function has nothing to do with zero-shot AD. Rather, the optimal loss is only related to the mixture weight $\pi$ during training. Without loss of generality, suppose we have two training distributions $P_1, P_2$. We learn a Deep SVDD model parameterized by $\theta$ and $c$ by the meta outlier exposure method,

$$
\begin{aligned}
&l(\theta, c) \\
&= \mathbb{E}_{x \sim P_1^\pi}\left[(1 - y_1)(f_\theta(x) - c)^2 + \frac{y_1}{(f_\theta(x) - c)^2}\right] + \mathbb{E}_{x \sim P_2^\pi}\left[(1 - y_2)(f_\theta(x) - c)^2 + \frac{y_2}{(f_\theta(x) - c)^2}\right] \\
&= \mathbb{E}_{x_1 \sim P_1, x_2 \sim P_2}\left[\pi(f_\theta(x_1) - c)^2 + \frac{1 - \pi}{(f_\theta(x_2) - c)^2} + \pi(f_\theta(x_2) - c)^2 + \frac{1 - \pi}{(f_\theta(x_1) - c)^2}\right] \quad (17) \\
&= \sum_{i=1}^{2} \mathbb{E}_{x_i \sim P_i}\left[\pi(f_\theta(x_i) - c)^2 + \frac{1 - \pi}{(f_\theta(x_i) - c)^2}\right] \\
&\geq 4\sqrt{\pi(1 - \pi)}
\end{aligned}
$$

where $0.5 < \pi < 1$ implies the majority assumption and the equality holds when the model parameters ($\theta$ and $c$) are tuned such that $(f_\theta(x_i) - c)^2 = \sqrt{(1 - \pi)/\pi}$ for any $x_i$. All data points will be put at the hypersphere's surface centered around $c$ with a radius $\sqrt{(1 - \pi)/\pi}$ in the feature space when the model is trivially optimized. However, the optimal loss has nothing to do with distinguishing different distribution's input data $x$ in the feature space, which is unlikely to produce useful representations for zero-shot AD.

On the other hand, if we apply batch normalization in the model $f_\theta$, we will not have the optimal loss function irrelevant to distributions. To see this, note that batch normalization will shift the input toward the origin. Thus $x_1$ in training task $P_1^\pi$ and $x_2$ in training task $P_2^\pi$ should have similar representations as they both take the majority in each task. Similarly, $x_2$ in task $P_1^\pi$ and $x_1$ in task $P_2^\pi$ are minorities, thus mapped far away from the origin in the feature space. Therefore, the symmetry breaks in Eq. (17), and the above trivial optimal loss disappears.

**Ablation Study on Batch Sizes.** To test the batch size effects, we add an ablation study on the Gaussian noise version of CIFAR-100C where we fix the anomaly ratio as 5% or 10% and try different batch sizes. The results are summarized in Tab. 6. It shows that larger batch sizes lead to more stable results. The performance is similar when the batch size is larger than or equal to 40. Even with a batch size being 20, our results are still better than the best-performing baseline.

We also test extreme batch sizes being 3, 6, 11, and 16 where each batch contains one anomaly.

**Ablation Study on Number of Training Classes.** To analyze the effect of the number of training distributions on the zero-shot AD performance, we conducted experiments on Omniglot where we varied the number of available meta-training classes from 1, 2, 5, 10, 15, 20, 40, 80 to 640, 1200. We separately trained ACR-DSVDD on each setup and tested the resulting models on the test set that has a 10% ground-truth anomaly ratio. We repeated 5 runs of the experiment with random initialization and reported the AUROC results in Tab. 7. It shows that using 320 available classes for this dataset is sufficient to achieve a decent zero-shot AD performance. The results also demonstrate that even though we have only one class in the meta-training set, thanks to the batch norm adaptation, we can still get better-than-random zero-shot AD performance.

**Ablation Study on Other Normalization Techniques.** As follows, we report new experiments involving LayerNorm, InstanceNorm, and GroupNorm for zero-shot AD.

We stress that, while these methods may have overall benefits in terms of enhancing performance, they do not work in isolation in our zero-shot AD setup. A crucial difference between these methods and batch normalization is that they treat each observation individually, rather than computing normalization statistics across a batch of observations. However, sharing information across the batch (and this way implicitly learning about distribution-level information) was crucial for our method to work.

Our experiments (AUROC results in the table below) with DSVDD on the Omniglot dataset support this reasoning. Using these normalization layers in isolations yields to random outcomes (AUROC=50):

| LayerNorm | InstanceNorm | GroupNorm |
|-----------|--------------|-----------|
| 50.0±0.9 | 50.6±0.7 | 50.2±0.5 |

We also added a version of the experiment where we combined these methods with batch normalization in the final layer. The results dramatically improve in this case:

| BatchNorm (BN) | LayerNorm + BN | InstanceNorm + BN | GroupNorm + BN |
|----------------|----------------|-------------------|----------------|
| 99.1±0.2 | 98.8±0.1 | 98.8±0.2 | 98.2±0.2 |

Experimental details: We use DSVDD as the anomaly detector and experiment on the Omniglot dataset. Each nonlinear layer of the feature extractor for DSVDD is followed by the respective normalization layer. We apply the same training protocol as Tab. 9 in the paper. For GroupNorm, we separate the channels into two groups wherever we apply group normalization.

**Effects of BatchNorm (BN) Layer Position.** We conducted additional experiments on two visual anomaly detection tasks – anomaly segmentation on the MVTec-AD dataset and object-level AD on CIFAR100C. We used the same DSVDD model architectures as used in Tables 1 and 2 as the backbone model, except that we switched off BN in all but one layer. For anomaly segmentation, there are five possible BN layer positions; and there are four positions for the object-level AD model. We switched off the BN layers in all but one position and then re-trained and tested the model with the same protocol used in our main paper (For CIFAR100C, we tested the model with the test data anomaly ratio of 10%). We iterate this procedure across all available BN layer positions. We repeat every experiment with different random seeds five times and report the mean AUROC and standard deviation. The results are summarized in the tables below, where a smaller value of the BN position corresponds to earlier layers (close to the input), and a larger value corresponds to later layers close to the output. The final column is copied from our results in Tables 1 and 2 where BN layers are on all available positions. For both MVTec-AD and CIFAR100C, we average the performance across all test classes.

Results on the two tasks have opposite trends regarding the effects of BN layer positions. Specifically, for anomaly segmentation on MVTec-AD, earlier BN layers are more effective, while for AD on CIFAR100C, later BN layers are more effective. This observation can be explained by the fact that anomaly segmentation is more sensitive to low-level features, while object-level AD is more sensitive to global feature representations. In addition, compared to the results in Tables 1 and 2 (copied to the last column in the table below), our results suggest that using BN layers at multiple positions does help re-calibrate the data batches of different distributions from low-level features (early layers) to high-level features (late layers) and shows performance improvement over a single BN layer.

| MVTec-AD | | | | | | |
|----------|---|---|---|---|---|---|
| BN Position | 1 | 2 | 3 | 4 | 5 | (1,2,3,4,5) |
| Pixel-level | 80.8±1.9 | 69.6±1.4 | 73.9±0.9 | 63.6±1.6 | 60.9±0.8 | 92.5±0.2 |
| Image-level | 74.7±0.9 | 59.2±1.6 | 63.6±1.3 | 65.5±1.2 | 65.4±1.3 | 85.8±0.6 |

| CIFAR100C | | | | | |
|-----------|---|---|---|---|---|
| BN Position | 1 | 2 | 3 | 4 | (1,2,3,4) |
| AUROC | 61.4±0.5 | 61.0±0.9 | 68.2±0.9 | 68.9±1.1 | 85.9±0.4 |

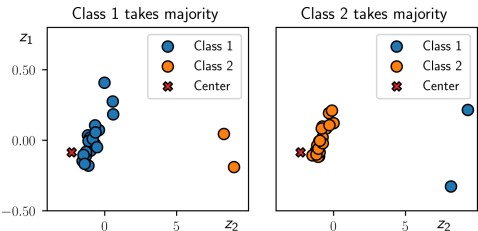

Figure 3: 2D visualization (after PCA) of the adaptively centered representations for two test tasks in the Omniglot dataset. The same learned DSVDD model adapts with our proposed method and maps samples from the majority class (class 1 (left) and class 2 (right)) to the same center in the embedding space in both tasks.

**Robustness of the mixing hyperparameter $\pi$ in Eq. (6).** We conduct the following experiments with varying $\pi$. The experiment has the same setup as Table 1 on CIFAR100C with a testing anomaly ratio of 0.1. The results show that all tested $\pi$'s results are over 84% AUC.

|  | CIFAR100C | | | | |
|---|---|---|---|---|---|
| $\pi$ | 0.99 | 0.95 | 0.9 | 0.8 | 0.6 |
| AUROC | 85.8±0.5 | 85.4±0.5 | 84.1±0.4 | 85.9±0.4 | 84.4±0.6 |

## I.2 Visualization of ACR.

We provide a visualization of the learned representations from DSVDD on the Omniglot dataset as qualitative evidence in Fig. 3. We observe that even though the normal and abnormal data classes flip in two plots, the model learns to center the samples from the majority class and map the samples from the minority class away to the center in the embedding space. In conclusion, ACR is an easy-to-use zero-shot AD method and achieves superior zero-shot AD results on different types of images. The performance of ACR is also robust against the test anomaly ratios.

## I.3 Additional Results on CIFAR100-C

We test all methods on all corruption types of CIFAR100-C. The results are presented in Tab. 8.

## I.4 Additional Results on Non-natural Images

**Datasets.** We further evaluate the methods on two other non-natural datasets–MNIST and Omniglot of hand-written characters. MNIST uses the same split and evaluation protocol as OrganA. On Omniglot, we take the first 1200 classes to form the meta-training set and use the remaining unseen 423 classes for testing.

**Results.** We present the results in Tab. 9. It shows that our approach significantly outperforms all the other baselines by a large margin on both datasets.

## I.5 Class-wise Results on MVTec-AD

We present the class-wise results in Tab. 10 for finer comparisons with other methods on MVTec AD benchmark.

Besides, we also implement the synthetic anomalies using images from different distributions during training. The result is worse than the model using Gaussian corrupt noise as example anomalies but still better than existing works in anomaly segmentation. We summarize the results in Tab. 11, which suggests that using images from different distributions as example anomalies during training is helpful for anomaly segmentation on MVTec-AD.

## I.6 Additional Results on Malware

**Dataset.** Malware [34] is a dataset of malicious and benign computer programs, collected from 11/2010 to 07/2014. Malware attacks are designed adversarially, thus leading to shifts in both normal

and abnormal data. We adopt the data reader from Li et al. [43]. We follow the preprocessing of [34] and convert the real-valued probabilities $p$ of being malware to binary labels (labeled one if $p > 0.6$ and zero if $p < 0.4$). The samples with probabilities between $0.4$ and $0.6$ are discarded. The model is trained on normal samples collected from 01/2011 to 12/2013, validated on normal and abnormal samples from 11/2010 to 12/2010, and tested on normal and abnormal samples from 01/2014 to 07/2014 (the anomaly ratios vary between $1\%$ and $20\%$).

**Results.** We report the results on Malware in Tab. 12. ACR-NTL achieves the best results under all anomaly ratios. All baselines except ICL perform worse than random guessing, meaning that the malware successfully fools most baselines, which testifies to the adversarial-upgrade explanation in the main paper.

Table 8: AUC (%) with standard deviation for anomaly detection on CIFAR100-C [29].

| Noise Type | Method | 1% | 5% | 10% | 20% |
|---|---|---|---|---|---|
| gaussian noise | ACR-DSVDD | 87.7±1.4 | 86.3±0.9 | 85.9±0.4 | 85.6±0.4 |
| | ACR-BCE | 84.3±2.2 | 86.0±0.3 | 86.0±0.2 | 85.7±0.4 |
| | ResNet152-I | 75.6±2.3 | 73.2±1.3 | 73.2±0.8 | 69.9±0.6 |
| | ResNet152-II | 62.5±3.1 | 61.8±1.7 | 61.2±0.6 | 60.2±0.4 |
| | OC-MAML (1-shot) | 53.0±3.6 | 54.1±1.9 | 55.8±0.6 | 57.1±1.0 |
| | CLIP-AD | 82.3±1.1 | 82.6±0.9 | 82.3±0.9 | 82.6±0.1 |
| shot noise | ACR-DSVDD | 85.5±1.6 | 86.5±0.2 | 87.3±0.6 | 86.4±0.4 |
| | ACR-BCE | 87.1±2.4 | 86.3±0.6 | 86.8±0.5 | 86.4±0.1 |
| | ResNet152-I | 76.9±2.3 | 75.7±0.7 | 74.3±0.6 | 71.9±0.6 |
| | ResNet152-II | 59.7±2.0 | 60.9±1.4 | 61.0±0.6 | 60.1±0.6 |
| | OC-MAML (1-shot) | 53.8±4.7 | 52.8±1.1 | 53.6±1.0 | 53.8±1.3 |
| | CLIP-AD | 83.0±1.6 | 84.1±0.3 | 83.9±0.5 | 83.3±0.3 |
| impulse noise | ACR-DSVDD | 80.5±3.7 | 81.5±0.5 | 80.7±0.7 | 79.8±0.2 |
| | ACR-BCE | 81.7±1.0 | 81.0±0.5 | 80.8±0.7 | 79.5±0.3 |
| | ResNet152-I | 74.3±1.4 | 73.1±1.0 | 72.2±0.4 | 69.4±0.3 |
| | ResNet152-II | 64.3±2.7 | 63.0±1.2 | 62.2±0.8 | 61.2±0.6 |
| | OC-MAML (1-shot) | 53.6±2.5 | 54.8±1.6 | 53.6±1.1 | 53.8±0.9 |
| | CLIP-AD | 81.5±2.0 | 82.7±0.4 | 82.3±0.5 | 82.2±0.2 |
| speckle noise | ACR-DSVDD | 86.5±2.0 | 85.8±0.8 | 86.0±0.4 | 85.1±0.2 |
| | ACR-BCE | 85.9±1.7 | 86.4±0.4 | 85.7±0.6 | 85.4±0.4 |
| | ResNet152-I | 75.8±2.8 | 75.8±0.4 | 75.1±0.4 | 72.9±0.5 |
| | ResNet152-II | 61.8±2.8 | 61.0±1.0 | 61.0±0.9 | 59.8±0.3 |
| | OC-MAML (1-shot) | 52.2±2.7 | 52.8±1.2 | 53.5±1.2 | 53.7±0.4 |
| | CLIP-AD | 84.6±1.6 | 83.7±0.4 | 84.1±0.4 | 84.2±0.3 |
| gaussian blur | ACR-DSVDD | 88.5±1.1 | 88.5±0.7 | 88.7±0.4 | 88.6±0.3 |
| | ACR-BCE | 85.6±1.3 | 85.0±0.6 | 85.0±0.9 | 84.7±0.5 |
| | ResNet152-I | 85.2±1.5 | 83.7±1.0 | 82.9±0.7 | 80.9±0.3 |
| | ResNet152-II | 64.9±1.5 | 65.3±1.2 | 64.0±0.9 | 62.7±0.4 |
| | OC-MAML (1-shot) | 55.6±3.6 | 56.6±0.6 | 56.8±1.1 | 57.6±0.6 |
| | CLIP-AD | 91.9±0.8 | 92.7±0.5 | 92.1±0.5 | 92.3±0.2 |
| defocus blur | ACR-DSVDD | 89.7±1.8 | 89.5±0.8 | 89.1±0.3 | 89.2±0.3 |
| | ACR-BCE | 86.5±1.3 | 86.5±0.6 | 86.3±0.3 | 85.9±0.4 |
| | ResNet152-I | 85.9±2.3 | 85.5±0.8 | 83.8±0.6 | 82.4±0.3 |
| | ResNet152-II | 66.0±1.8 | 65.4±1.1 | 63.7±0.6 | 63.2±0.3 |
| | OC-MAML (1-shot) | 53.5±2.5 | 51.7±1.7 | 54.0±1.8 | 54.7±0.7 |
| | CLIP-AD | 93.1±1.4 | 92.9±0.3 | 92.8±0.3 | 92.8±0.2 |
| glass blur | ACR-DSVDD | 87.0±2.1 | 87.9±0.4 | 87.7±0.4 | 87.6±0.2 |
| | ACR-BCE | 85.4±1.0 | 86.1±0.4 | 86.4±0.4 | 86.1±0.3 |
| | ResNet152-I | 80.3±2.5 | 78.7±0.8 | 78.0±0.6 | 75.6±0.4 |
| | ResNet152-II | 63.9±2.2 | 63.0±1.7 | 63.6±0.5 | 62.2±0.4 |
| | OC-MAML (1-shot) | 52.8±1.9 | 53.1±1.7 | 53.9±0.9 | 53.7±1.4 |
| | CLIP-AD | 85.4±0.5 | 85.0±1.1 | 84.2±0.7 | 84.4±0.3 |
| motion blur | ACR-DSVDD | 89.2±0.4 | 89.6±0.8 | 89.1±0.6 | 88.6±0.5 |
| | ACR-BCE | 86.3±1.9 | 85.3±1.0 | 85.7±0.2 | 84.9±0.2 |
| | ResNet152-I | 84.3±1.3 | 83.4±1.3 | 82.0±0.5 | 80.4±0.3 |
| | ResNet152-II | 66.6±3.1 | 64.8±1.2 | 63.4±0.6 | 62.4±0.3 |
| | OC-MAML (1-shot) | 50.5±3.1 | 52.3±1.6 | 53.1±0.9 | 53.6±0.7 |
| | CLIP-AD | 91.8±1.4 | 92.9±0.4 | 92.7±0.3 | 92.8±0.3 |
| zoom blur | ACR-DSVDD | 90.3±1.7 | 89.6±0.7 | 89.8±0.4 | 89.4±0.3 |
| | ACR-BCE | 87.5±1.8 | 86.5±1.0 | 86.4±0.2 | 86.4±0.3 |
| | ResNet152-I | 86.9±1.3 | 87.2±0.3 | 86.3±0.3 | 84.6±0.2 |
| | ResNet152-II | 65.2±1.1 | 65.7±0.7 | 66.1±0.3 | 64.2±0.4 |
| | OC-MAML (1-shot) | 50.6±3.2 | 53.8±0.8 | 53.7±1.4 | 54.2±0.4 |
| | CLIP-AD | 94.2±1.4 | 94.4±0.3 | 94.3±0.3 | 93.9±0.3 |
| snow | ACR-DSVDD | 87.7±1.2 | 87.7±1.0 | 87.6±0.4 | 87.4±0.3 |
| | ACR-BCE | 84.4±2.6 | 85.5±0.8 | 85.5±0.6 | 84.4±0.0 |
| | ResNet152-I | 85.8±1.4 | 84.8±0.6 | 83.7±0.8 | 81.9±0.3 |
| | ResNet152-II | 67.1±1.9 | 65.6±0.9 | 64.5±0.4 | 63.3±0.8 |
| | OC-MAML (1-shot) | 56.7±4.5 | 54.5±1.8 | 56.8±0.6 | 57.3±0.2 |
| | CLIP-AD | 91.7±0.8 | 92.9±0.4 | 93.3±0.2 | 93.2±0.2 |
| fog | ACR-DSVDD | 86.2±1.8 | 85.2±0.6 | 85.4±0.9 | 85.0±0.2 |
| | ACR-BCE | 78.8±2.7 | 77.7±0.5 | 77.3±0.7 | 77.2±0.6 |
| | ResNet152-I | 76.4±1.8 | 76.9±0.6 | 74.8±1.0 | 73.0±0.9 |
| | ResNet152-II | 64.5±2.1 | 62.9±0.8 | 62.5±0.4 | 61.0±0.5 |
| | OC-MAML (1-shot) | 51.9±3.6 | 52.9±0.9 | 53.4±0.6 | 53.7±0.2 |
| | CLIP-AD | 91.9±0.8 | 92.3±0.5 | 92.2±0.4 | 92.3±0.3 |
| frost | ACR-DSVDD | 88.2±1.5 | 88.0±0.9 | 87.4±0.6 | 87.2±0.3 |
| | ACR-BCE | 83.2±1.4 | 84.1±1.2 | 84.6±0.6 | 83.7±0.4 |
| | ResNet152-I | 85.9±1.6 | 85.5±0.5 | 83.8±0.8 | 81.4±0.5 |
| | ResNet152-II | 63.0±1.0 | 63.2±0.5 | 62.7±1.3 | 61.7±0.3 |
| | OC-MAML (1-shot) | 52.8±1.3 | 52.4±2.0 | 53.6±0.7 | 53.2±1.1 |
| | CLIP-AD | 92.9±0.6 | 93.1±0.2 | 93.6±0.3 | 93.2±0.2 |
| brightness | ACR-DSVDD | 90.0±1.5 | 89.5±0.9 | 89.6±0.4 | 89.9±0.2 |
| | ACR-BCE | 86.7±1.3 | 87.8±0.7 | 87.1±0.8 | 87.2±0.4 |
| | ResNet152-I | 90.7±0.9 | 90.8±0.5 | 89.7±0.3 | 88.1±0.3 |
| | ResNet152-II | 67.6±2.1 | 69.8±0.4 | 68.2±1.0 | 67.0±0.5 |
| | OC-MAML (1-shot) | 53.6±1.1 | 56.8±1.5 | 56.2±0.7 | 56.8±0.5 |
| | CLIP-AD | 94.6±0.4 | 95.6±0.3 | 95.4±0.3 | 95.3±0.2 |
| spatter | ACR-DSVDD | 88.1±1.5 | 89.2±0.6 | 89.0±0.6 | 88.7±0.1 |
| | ACR-BCE | 86.2±2.3 | 87.7±0.3 | 87.2±0.6 | 87.3±0.3 |
| | ResNet152-I | 90.6±1.2 | 90.2±0.5 | 89.8±0.3 | 87.9±0.3 |
| | ResNet152-II | 68.7±1.7 | 67.6±0.9 | 66.0±0.9 | 65.2±0.4 |
| | OC-MAML (1-shot) | 53.6±2.7 | 55.6±1.1 | 56.1±0.7 | 53.6±1.5 |
| | CLIP-AD | 94.7±0.6 | 95.2±0.4 | 95.1±0.2 | 95.0±0.3 |
| saturate | ACR-DSVDD | 88.1±2.1 | 87.1±0.7 | 87.1±0.5 | 85.8±0.4 |
| | ACR-BCE | 86.8±2.0 | 86.1±0.8 | 86.0±0.6 | 85.3±0.3 |
| | ResNet152-I | 90.4±0.9 | 89.7±0.7 | 89.2±0.5 | 87.4±0.2 |
| | ResNet152-II | 67.7±1.8 | 67.7±1.4 | 67.4±0.8 | 65.9±0.3 |
| | OC-MAML (1-shot) | 55.6±2.4 | 53.5±0.9 | 55.1±0.8 | 54.1±1.2 |
| | CLIP-AD | 94.7±0.8 | 94.7±0.2 | 95.0±0.1 | 95.1±0.2 |
| contrast | ACR-DSVDD | 76.4±1.8 | 75.1±1.8 | 74.9±0.5 | 74.5±0.4 |
| | ACR-BCE | 67.6±2.0 | 66.7±0.8 | 67.8±0.7 | 66.9±0.3 |
| | ResNet152-I | 76.1±1.6 | 77.0±0.8 | 75.2±0.3 | 73.5±0.2 |
| | ResNet152-II | 61.3±0.9 | 61.3±1.2 | 60.2±0.5 | 59.3±0.5 |
| | OC-MAML (1-shot) | 54.6±3.7 | 54.0±0.3 | 53.1±1.2 | 54.1±1.0 |
| | CLIP-AD | 89.3±1.8 | 88.9±0.5 | 88.3±0.4 | 88.8±0.2 |
| elastic transform | ACR-DSVDD | 90.8±1.9 | 89.3±0.7 | 90.0±0.4 | 89.3±0.3 |
| | ACR-BCE | 87.6±1.0 | 86.7±0.8 | 87.4±0.6 | 87.2±0.4 |
| | ResNet152-I | 82.4±2.4 | 80.9±0.4 | 80.0±0.9 | 78.4±0.2 |
| | ResNet152-II | 65.6±2.3 | 65.2±0.6 | 63.9±0.7 | 62.0±0.3 |
| | OC-MAML (1-shot) | 52.5±3.9 | 54.3±1.2 | 54.4±1.2 | 54.7±0.8 |
| | CLIP-AD | 89.1±1.1 | 90.0±0.3 | 89.4±0.5 | 89.4±0.3 |
| pixelate | ACR-DSVDD | 91.7±0.5 | 91.1±0.6 | 90.8±0.6 | 90.7±0.2 |
| | ACR-BCE | 89.6±1.9 | 89.9±0.6 | 89.7±0.1 | 89.8±0.3 |
| | ResNet152-I | 82.5±1.6 | 83.2±1.3 | 82.5±0.7 | 80.1±0.3 |
| | ResNet152-II | 66.4±1.5 | 65.6±0.6 | 64.9±0.6 | 63.8±0.3 |
| | OC-MAML (1-shot) | 56.4±3.8 | 55.8±0.9 | 56.4±0.7 | 57.0±0.9 |
| | CLIP-AD | 86.7±0.3 | 86.7±0.7 | 86.9±0.3 | 86.7±0.3 |
| jpeg compression | ACR-DSVDD | 89.8±1.4 | 91.0±0.5 | 90.5±0.7 | 90.4±0.3 |
| | ACR-BCE | 89.1±1.2 | 88.8±0.8 | 89.1±0.5 | 88.6±0.3 |
| | ResNet152-I | 84.7±2.1 | 85.8±1.1 | 84.4±0.8 | 82.7±0.2 |
| | ResNet152-II | 62.9±2.4 | 63.9±0.8 | 63.0±0.8 | 61.3±0.8 |
| | OC-MAML (1-shot) | 52.0±2.7 | 55.6±0.9 | 56.4±1.1 | 57.2±1.8 |
| | CLIP-AD | 89.8±1.9 | 87.7±0.1 | 88.3±0.3 | 88.5±0.3 |

Table 9: AUC (%) with standard deviation for anomaly detection on non-natural images: Omniglot, MNIST, and OrganA. ACR with both backbone models outperforms all baselines on all datasets. In comparison, CLIP-AD performs much worse on non-natural images.

| | MNIST | | | Omniglot | | |
|---|---|---|---|---|---|---|
| | 1% | 5% | 10% | 5% | 10% | 20% |
| ADIB [14] | 50.4±2.0 | 49.4±1.7 | 49.4±2.0 | 50.8±1.7 | 49.5±0.6 | 49.7±0.4 |
| ResNet152-I [27] | 87.2±1.3 | 84.2±0.2 | 80.9±0.2 | 96.4±0.4 | 95.5±0.3 | 94.3±0.2 |
| ResNet152-II [27] | 80.0±1.9 | 78.4±1.5 | 74.9±0.3 | 88.1±0.8 | 86.7±0.5 | 84.4±0.6 |
| OC-MAML [22] | 83.7±3.5 | 86.0±2.3 | 86.4±2.8 | 98.6±0.3 | 98.4±0.2 | 98.5±0.1 |
| CLIP-AD [51] | 53.9±1.4 | 53.7±0.9 | 53.9±0.8 | N/A | N/A | N/A |
| ACR-DSVDD | **91.9±0.8** | **90.4±0.2** | **88.8±0.2** | **99.1±0.2** | **99.1±0.2** | **99.2±0.0** |
| ACR-BCE | 88.7±0.6 | 87.8±0.4 | 86.5±0.3 | 98.5±0.2 | **98.9±0.1** | **99.1±0.1** |

Table 10: ACR-DSVDD's pixel-level (segmentation) and image-level (classification) AUCROCs (%) on MVTec-AD.

| | Pixel-level | Image-level |
|---|---|---|
| Bottle | 95.8±0.5 | 99.5±0.2 |
| Cable | 87.1±1.1 | 72.2±1.9 |
| Capsule | 95.2±0.9 | 78.1±1.2 |
| Carpet | 97.8±0.4 | 99.8±0.2 |
| Grid | 90.4±1.4 | 83.8±3.0 |
| Hazelnut | 92.3±0.7 | 79.2±0.9 |
| Leather | 98.6±0.1 | 100.0±0.0 |
| Metal-nut | 79.5±2.5 | 75.6±5.4 |
| Pill | 93.0±1.3 | 72.2±3.2 |
| Screw | 86.7±0.9 | 48.5±3.1 |
| Tile | 90.3±0.9 | 98.4±0.3 |
| Toothbrush | 98.2±0.1 | 97.0±0.9 |
| Transistor | 97.2±0.1 | 93.1±1.1 |
| Wood | 89.2±1.1 | 98.2±0.8 |
| Zipper | 95.8±0.9 | 90.7±0.9 |
| Average | 92.5±0.2 | 85.8±0.6 |

Table 11: ACR-DSVDD's pixel-level (segmentation) and image-level (classification) AUCROCs (%) on MVTec-AD. The model uses images from other classes as synthetic anomalies during training.

| | Pixel-level | Image-level |
|---|---|---|
| Bottle | 94.5 | 98.6 |
| Cable | 88.1 | 64.5 |
| Capsule | 90.1 | 70.8 |
| Carpet | 97.5 | 99.5 |
| Grid | 74.8 | 97.9 |
| Hazelnut | 84.3 | 61.9 |
| Leather | 97.5 | 99.1 |
| Metal-nut | 67.5 | 54.2 |
| Pill | 89.0 | 66.8 |
| Screw | 76.6 | 53.2 |
| Tile | 90.0 | 97.6 |
| Toothbrush | 93.5 | 80.8 |
| Transistor | 95.2 | 91.4 |
| Wood | 88.1 | 97.2 |
| Zipper | 78.6 | 79.7 |
| Average | 87.0 | 78.8 |

Table 12: AUC (%) with standard deviation for anomaly detection on Malware [34]. ACR-NTL achieves the best results on various anomaly ratios.

|  | 1% | 5% | 10% | 20% |
|---|---|---|---|---|
| OC-SVM | 19.5±5.6 | 20.5±1.4 | 20.3±0.9 | 20.3±0.8 |
| IForest | 22.8±2.9 | 22.9±1.2 | 23.3±0.6 | 23.4±0.8 |
| LOF | 22.3±4.9 | 23.2±1.8 | 23.3±1.3 | 23.2±0.4 |
| KNN | 21.6±6.3 | 22.5±1.6 | 22.7±0.9 | 22.6±0.9 |
| DSVDD | 25.4±3.3 | 27.4±1.7 | 28.9±0.9 | 28.3±0.8 |
| AE | 48.8±2.4 | 49.1±1.2 | 49.4±0.6 | 49.3±0.5 |
| LUNAR | 23.1±4.5 | 23.8±1.2 | 24.1±0.7 | 24.2±0.6 |
| ICL | 83.5±1.9 | 81.0±1.0 | 82.9±0.8 | 83.1±0.9 |
| NTL | 25.9±4.8 | 25.4±1.3 | 24.5±1.3 | 25.0±0.8 |
| ACR-DSVDD | 73.1±2.8 | 69.5±3.3 | 69.4±3.3 | 66.4±4.0 |
| ACR-NTL | **85.0±1.3** | **84.5±0.8** | **85.1±1.2** | **84.0±0.8** |

