# OpenReview forum: "Zero-Shot Anomaly Detection via Batch Normalization"
_NeurIPS.cc/2023/Conference — NeurIPS 2023 poster_

### Official Review · Reviewer_2urY · 2023-06-18

**Soundness:** 3 good
**Presentation:** 2 fair
**Contribution:** 2 fair
**Rating:** 5
**Confidence:** 4

**Summary:**

The paper presents an approach to address the challenging task of zero-shot anomaly detection. The authors base their methodology on three crucial assumptions: the presence of a training set that shares relevant structures with the target task, performing batch-level analysis during inference, and having the majority of the batch comprised of normal class data. Leveraging these assumptions, the authors propose a method that effectively utilizes batch statistics to assign anomaly scores.
The proposed method, named Adaptive Centered Representations (ACR), combines batch normalization with meta-training. Through re-centering and scaling the normal samples within a batch, ACR successfully brings them closer to the center while pushing anomalies further away. Experiments were conducted in the study across different types of data, including both image and tabular data.


**Strengths:**

The authors tackle the task of zero-shot anomaly detection, which holds significant practical importance in various domains. The clear and well-written notation and assumption section greatly contributes to understanding the authors' approach and provides a solid foundation for the rest of the paper. Experimental evaluations conducted demonstrate the effectiveness of the proposed approach.



**Weaknesses:**

The paper has some weaknesses that should be addressed. Mainly, the reliance on some critical assumptions, such as the availability of a meta-training set and batch-level anomaly detection, may limit the practical applicability of the proposed method.

1. The assumption of a meta-training set assumes strong prior knowledge about the types of anomalies, which may not be feasible or practical in many real-world scenarios.

2. Table 7 shows that strong performance required over 100 different classes during training. This raises concerns about the scalability and feasibility of the method when faced with a smaller number of training classes. This is indicated by the lower performance observed when classes were reduced to 20. Further exploration of the method's performance with a smaller number of training sets would provide valuable insights into its limitations and generalization capabilities.

3. The experimental section of the paper could benefit from improved clarity and organization. It was challenging to follow the exact details of the conducted experiments, requiring multiple passes through the main paper and the Appendix. Enhancing the clarity and structure of the experimental section would greatly improve reproducibility and understanding of the reported results.


**Questions:**

1. The paper primarily addresses the one-to-many anomaly detection scenario, but it would be interesting to evaluate its performance in the more realistic many-vs-one scenario [1]. In this scenario, where the data is multi-modal and achieving compactness is more challenging. How the proposed method works on this setting?

2. The assumption of batch-level inference may not be applicable in many practical scenarios, such as the mentioned hospitals and clinics where doctors often assess patients individually rather than in large batches. The performance drop observed in Table 6 with a batch size of 20 raises questions about the method's effectiveness with even smaller batch sizes, such as 5 or 10. What is the performance of the proposed method with lower batch sizes?


References:
[1] Ahmed, F.; and Courville, A. Detecting semantic anomalies. AAAI 2020.

**Limitations:**

Yes.

---

> ### Author Rebuttal · Authors · 2023-08-09
>
> > The assumption of a meta-training set is strong
>
> This assumption is common in the meta-training literature, both for general methods, such as MAML as well as meta-training AD work [8, 21, 31, 39]. There are multiple ways to mitigate this assumption. If one does not have a meta-training set at hand, one can train their model on a different but related dataset, .e.g, train on Omniglot but test on MNIST (see results below under this setting. We still get decent AUC results on MNIST).
>
> Anomaly ratio|1%|5%|10%|20%|
> |:---|---|---|---|---|
> |AUROC|84.4$\pm$2.4|85.2$\pm$2.5|84.3$\pm$2.5|82.2$\pm$2.4|
> Rebuttal Table 4: MNIST
>
> In our main paper, we provide several examples of how to construct a meta-training set for real applications: for our tabular data benchmark, we use the timestamps to construct a meta-training set; in our MVTec-AD experiment, we simply use multiple industrial object classes to construct the meta-training set; in medical domains, we have data from different hospitals or patients or organs to construct the meta-training set.
>
> > Experiments with the extreme number of classes during training (e.g., 1)
>
> Thank you for mentioning this extreme setting. We will add the ablation study to Table 7. The results show that even though we have only one class in the meta-training set, thanks to the batch norm adaptation, we can still get better-than-random zero-shot AD performance.
>
> |#Training classes|1|2|5|10|15|
> |:---|---|---|---|---|---|
> |AUROC|59.0$\pm$0.6|71.8$\pm$0.6|72.5$\pm$0.3|72.2$\pm$1.0|75.3$\pm$0.4|
> Rebuttal Table 5
>
> > Experimental section lacks clarity and organization.
>
> We did multiple experiments on various data types and anomaly detection tasks. Our experimental section is structured by data types. For each data type, we first provide implementation details (train/val/test split and meta-training/testing construction) and then the results. In addition, pseudo-code and our codebase are provided in the supplement for reproducibility. We are grateful about more specific advice that we could incorporate.
>
> > How the proposed method works in other settings?
>
> Many-vs-one is the reversed version of the one-vs-rest setting. We study a more realistic setup: the N-vs-rest setting on Omniglot. We train the model under the one-vs-rest setup but test under multiple normal modes. The table below shows that as the number of normal classes (N) increases, the performance decreases because the sample mean and variance are not representative of all normal classes. We experiment with the anomaly ratio being 0.1 at test time.
>
> |N=1|N=2|N=5|N=10|N=20|
> |---|---|---|---|---|
> |99.0|95.6|81.5|65.1|28.9|
> Rebuttal Table 6
>
> > Validity of batch-level anomaly detection.
>
> Batch-level predictions are widely used in real life. For example, people examine Covid19 test samples at a batch level out of economic and time-efficiency considerations (https://www.fda.gov/medical-devices/coronavirus-covid-19-and-medical-devices/pooled-sample-testing-and-screening-testing-covid-19). Our work makes similar assumptions as batch-level predictions in the literature [10, 47, 51, 64, 73]. Our method can be easily extended to score individual data by presetting the sample mean and variance in BatchNorm layers with a collection of data. These moments are then fixed when predicting new individual data.
>
>
> > Performance in lower batch sizes.
>
> The table below shows the performance in the lower batch size regime. We will add the results below to Table 6. Each batch contains 1 anomaly.
>
> |Batch size|3|6|11|16|
> |:---|---|---|---|---|
> |AUROC|66.4 $\pm$ 2.3|77.9 $\pm$ 2.8|82.3 $\pm$ 2.7|84.8 $\pm$ 2.0|
> Rebuttal Table 7

---

> > ### Comment · Reviewer_2urY · 2023-08-17
> >
> > Thanks for providing detailed results and for responding to my concerns. I have read the author's responses and all other reviews, but I am keeping my rating the same.

---

> > > ### Author Response · Authors · 2023-08-18
> > > **Thank you for your time in reviewing our paper**
> > >
> > > Thank you for your time in reviewing our paper! We appreciate your feedback.

---

### Official Review · Reviewer_GN5r · 2023-07-01

**Soundness:** 2 fair
**Presentation:** 3 good
**Contribution:** 2 fair
**Rating:** 5
**Confidence:** 4

**Summary:**

This paper proposes a method called Adaptive Centered Representations (ACR) for zero-shot batch-level anomaly detection. The method utilizes off-the-shelf deep anomaly detectors trained on a meta-training set, combined with batch normalization layers, to achieve automatic zero-shot generalization for unseen anomaly detection tasks. The main contributions are introducing an effective method for zero-shot anomaly detection, demonstrating its applicability to tabular data, and achieving competitive results in anomaly segmentation on specialized image datasets.




**Strengths:**

In this paper, the effectiveness of the proposed method is verified by a large number of experiments. There is a rigorous mathematical proof process, and the effectiveness of the method is verified by experiments on multiple data sets
The results are powerful. In most cases, the proposed methods perform very well.

**Weaknesses:**

1.The article claims that Batch Normalization Layers can improve the model's adaptive ability in new tasks. In article 4.1.1, Anomaly Detection experiment of CIFAR100-C with Gaussian noise and medical image dataset OrganA was conducted. Why ACR-DSVDD with the addition of Batch Normalization Layers performs better overall than ACR-BCE is not further explained here.

2.As for the anomaly ratio π in this paper, I think it is also a major hyperparameter indicator. The paper only claims that the model is robust when it is equal to 0.01, 0.05, 0.1, 0.2, and 0.4. Ablation experiments with this parameter are expected to prove its robustness. In the Implementation Details of section 4.1.1 it is claimed that π=0.8, but the supplementary material says that π=0.2

**Questions:**

1. In order to improve the clarity and understandability of this paper, it is suggested to add a schematic diagram outlining the overall method. The visualization diagram of the whole paper is too little, which makes it difficult to grasp the overall method principle.
2. When discussing meta-learning, I want to explain some basic concepts, rather than explaining Batch Normalization Layers, and I want to explain the pseudo-code of the supplementary material in the article, which ensures that the reader has a full understanding of the background and significance of these models.

**Limitations:**

Limitations are clearly described and I don't expect negative societal impact of this work.

---

> ### Author Rebuttal · Authors · 2023-08-09
>
> > Why ACR-DSVDD with the addition of Batch Normalization Layers performs better overall than ACR-BCE is not further explained here.
>
> DSVDD has a specific inductive bias for anomaly detection to learn a compact normal data boundary, while a binary classifier doesn’t have such an inductive bias. Li et al. 2023 visualize this inductive bias through the normal data boundary in Fig 1, showing that a binary classifier using binary cross entropy loss (BCE) has a more conservative normal data boundary than DSVDD.
>
> [Li et al. Deep Anomaly Detection under Labeling Budget Constraints. ICML 2023.]
>
> > Robustness of the mixing hyperparameter $\pi$ in Eq. 6.
>
> Thanks for mentioning this ablation study. We conduct the following experiments with varying $\pi$. The experiment has the same setup as Table 1 on CIFAR100C with a testing anomaly ratio of 0.1. The results show that all tested $\pi$’s results are over 84% AUC.
>
> |0.99|0.95|0.9|0.8|0.6|
> |---|---|---|---|---|
> |85.8$\pm$0.5|85.4$\pm$0.5|84.1$\pm$0.4|85.9$\pm$0.4|84.4$\pm$0.6|
> Rebuttal Table 3
>
> > Inconsistency values for the mixing hyperparameter $\pi$ in the main paper and the supplement.
>
> Thank you for noticing this inconsistency and we will correct the typos in the supplement. We report 1-$\pi$ in the supplement but report $\pi$ in the main paper. This will be fixed.

---

> > ### Comment · Reviewer_GN5r · 2023-08-17
> >
> > Thank you for answering my opinion. I am happy to stand by my original decision to recommend acceptance of the paper. I might want to cause you to attention to adding a diagram outlining the overall approach to the essay, which will help the reader understand your essay and ensure quality.

---

> > > ### Author Response · Authors · 2023-08-18
> > > **Thank you for recommending acceptance of our paper**
> > >
> > > Thank you for recommending acceptance of our paper! We will consider adding a diagram to explain our overall method in our revised version.

---

### Official Review · Reviewer_hQFt · 2023-07-01

**Soundness:** 3 good
**Presentation:** 4 excellent
**Contribution:** 3 good
**Rating:** 6
**Confidence:** 5

**Summary:**

This paper proposes Adaptive Centered Representations (ACR) for zero-shot batch-level AD. The simple recipe, batch normalization plus meta-training, seems highly effective and versatile. The experiments are well conducted on multiple datasets.

**Strengths:**

1.	This paper is well-written, and very easy to follow.
2.	The contributions are very clear.
3.	The experiments are well conducted on multiple datasets.

**Weaknesses:**

The main issue with this paper is the confusion between the problems of anomaly detection and one-class classification, which leads to several problems, including uncertainty and irrationality in experimental settings, unfair comparisons with competing methods, and misleading statements in the related works section.

1.	The key problem: The key problem lies in the authors' confusion between one-class classification and anomaly detection. In the context of industrial anomaly detection using datasets like MVTec, the training set consists of normal objects without any anomalies, while the testing phase targets specific fine-grained anomalies. This discrepancy makes the task extremely challenging. However, in one-class classification, particularly during meta-training, anomalies can be defined as any coarse-grained objects as they are with the same definition of the anomalies for the test, and the training set can include multiple objects, providing some form of supervision during meta-training.
2.	Misleading in the related works: There is confusion in the related works section, specifically in lines 194-206, where the authors misinterpret the settings of RegAD [31] compared to other papers [21, 8, 39]. While all these papers involve few-shot learning during meta-testing (although [31] does not explicitly position itself as meta-learning in its original paper), the crucial difference lies in the meta-training phase. In [21, 8, 39], data from multiple classes are used for supervised learning during the meta-training process for the one-class classification meta-task. However, in [31], the data from multiple objects cannot be used for supervised learning since the real anomalies are fine-grained defects, which do not exist in the entire meta-training dataset. As a result, the difficulty of the two tasks in the meta-training phase is fundamentally different, and the authors should clearly clarify these distinctions to avoid misleading readers.
3.	Uncertainty and irrationality of experimental settings: There are uncertainties and irrationalities in the experimental settings, particularly in the training process for defect detection on the MVTec dataset. It is unclear how anomalous data is included in the training set of the 14 classes, as the authors suggest that most of the data in one batch are normal. It would be helpful to clarify whether some anomalous data is included in the test set of these 14 classes for training purposes. Additionally, during testing, it is unclear how the test data batches are organized. If some classes in MVTec have more anomalous data than normal data, it becomes crucial to address how this imbalance is handled, as it deviates from the assumed scenario.
4.	Unfair comparisons with competing methods: Unfair comparisons are made with competing methods. The traditional definition of zero-shot learning implies that the results can be determined for each independent sample independently. In Table 2, methods such as WinCLIP [35] can classify individual images as normal or anomalous independently. However, in this paper, decisions can only be made after processing a large batch of data, which deviates from the true zero-shot setting as demonstrated in [35].

Overall, this is a good paper with clear contributions; however, it can mislead readers and create confusion regarding some important experimental settings.

**Questions:**

Please refer to the weaknesses.

---

> ### Author Rebuttal · Authors · 2023-08-09
>
> > relation between one-class classification and anomaly detection.
>
> We wonder whether our understanding of the question aligns with your intention, so let us give our understanding of the anomaly detection (AD) problem. AD is a broad field that can be formulated in many ways (Table 3 in Ruff et al. 2021). We consider two setups: image-level anomaly detection (the object-level one-vs-rest setup) and anomaly segmentation (featuring the MVTec-AD dataset). In both cases, we aim to distinguish observed data (considered normal) from distributionally-different data (anomalies) encountered in the future. One-class classification (OCC) is a widely-used method to solve this problem. In contrast to binary classification, OCC tries to learn a tight (e.g., circular) decision boundary around the observed normal class (See Fig. 1 in Li et al., 2023), thus distinguishing it from everything else encountered in the future. This is different from binary classification, which usually involves an unbounded (e.g., linear) decision boundary. We don’t expect the potential problems laid out in this question since our approach promotes robustness to future anomalies, as we ensure that anomalies and normal data encountered during testing (e.g., MNIST digits 5-9) are held out during meta-training (which in the MNIST example only uses digits 0-4).
>
> [Ruff, Lukas, et al. "A unifying review of deep and shallow anomaly detection." Proceedings of the IEEE 109.5 (2021): 756-795.]
>
> [Li et al., Deep Anomaly Detection under Labeling Budget Constraints, ICML 2023]
>
> > anomalies can be defined as any coarse-grained objects as they are with the same definition of the anomalies for the test, and the training set can include multiple objects, providing some form of supervision during meta-training.
>
> We are not sure we understand the question–are you suggesting that our approach may not be robust to novel types of anomalies encountered during testing? Even though we use a meta-training set, our train-test split of the classes ensures that the objects or classes in the meta-training set are not present at test time, so there is no “information leakage” between the data distributions encountered between meta-training and testing. The availability of a meta-training set provides the possibility of supervising the anomaly detector with synthetic abnormal data from other classes. Second, our ablation study showed that using the supervision signal from synthetic abnormal data at training time leads to better zero-shot AD performance than not using it (compare “one-class loss” in Table 4 against the results in Table 1).
>
> > Additional information on RegAD [31] in the related works
>
> Thanks for your information on the additional differences between RegAD [31] and other related works. We will clarify by adding the following sentence in the related work: “RegAD[31] does not exploit the presence of a meta-set to learn a stronger anomaly detector through synthetic outlier exposure. We found that using images from different distributions as example anomalies during training is helpful for anomaly segmentation on MVTec.”
>
> > Meta-training and testing details on MVTec
>
> The implementation details can be found in Section 4.2, E.2, and our codebase. In summary,
> * When we aim to detect anomalies in one targeted class, we construct the meta-training set by removing the target class from the original MVTec training set.
> * We use class-level images to form supervision signals during meta-training. We don’t use any realistic defects from the official test set at training time.
> * We ensure that we have a clean split between distributions (object classes) available for meta-training and meta-testing. This includes training and test sets.
> * Since our goal is anomaly segmentation, we assign an anomaly score to each data patch, using a sliding window approach.
> * Note that the assumption that normal data are more present than anomalies still holds for anomaly segmentation since anomalous segments are usually localized in tiny regions in the images. (We also noticed that at the image level, there could be more anomalous images than normal ones but this isn’t the case at the patch level.)
>
> > Unfair comparisons with competing methods
>
> We disagree that our usage of batch-level prediction is unfair to other baselines that score data individually. Zero-shot learning relies on auxiliary information to inform a model of what the current task is [Xian et al., 2018]. Zero-shot AD approaches operate differently when it comes to identifying the “new normal” distribution. WinCLIP carefully selects the prompts handmade by human experts as auxiliary information for normality. In contrast, we use batch-level information as auxiliary information to achieve zero-shot anomaly detection, for which we don’t need any human experts at all. The benefits of our method are three-fold. 1) Batches are cheaper than human expert labeling and prompt engineering, so there is no additional cost. 2) Our method also works for small batches in our ablation studies. 3) We can extend our method to score individual data by presetting the sample mean and variance in BatchNorm layers with a collection of data. These moments are then fixed when predicting anomaly scores for new individual data.
>
> [Xian, Yongqin, et al. “Zero-shot learning—a comprehensive evaluation of the good, the bad and the ugly. TPAMI 2018]

---

> > ### Comment · Reviewer_hQFt · 2023-08-10
> >
> > I greatly appreciate the authors' thoughtful response.
> >
> > To Q1 and Q2: Thank you for your comprehensive explanation. Please rest assured, as I have a clear understanding of the concept of anomaly detection. To emphasize the differences, in the context of One-Class Classification (OCC) using the MNIST dataset, models trained on digits 0-4, when with 0 as the norm, can identify digits 1-4 as anomalies. The meta-set teaches the model to recognize distinct digits based on their identity labels. When applied during testing, if digit 5 is designated as normal during testing, the model can accurately flag digits 6-9 as anomalies, without being troubled by potential variations such as "imperfectly written digit 5" or "blurred image of digit 5". This understanding is achieved through meta-training, allowing the model to grasp the fundamental anomaly types associated with different digit identities. even in cases where classes have no overlap. In contrast, in industrial anomaly detection on the MVTec dataset, anomalies are defects within the same object category. However, as the MVTec meta-training set lacks defective instances for any categories, the model lacks prior knowledge of potential anomaly types. Intuitively, using various defect-free images (like tiles or transistors) as training data doesn't adequately prepare the model for detecting nuanced defects within one category (e.g., identifying subtle defects in wood textures). Therefore, the latter task of MVTec is inherently more challenging than the OCC task.
> >
> > My intention isn't to raise doubts, but rather to highlight that the authors might not have explicitly differentiated between these tasks, potentially leading to a lessened emphasis on their distinct complexities. This could inadvertently cause readers to equate their difficulties, despite the inherent challenges in the MVTec task. I am happy to see that the authors already prepare to revise the related works to give a clear explanation. Could briefly summarize the differences in the above tasks and present them in the paper?
> >
> > To Q3: Could you clarify whether, during MVTec testing, a batch refers to patches within an image or the entire test dataset? Despite my careful review of the rebuttal and paper, this specific query remains unanswered.
> >
> > To Q4: I agree that using an entire batch for collective decision-making is reasonable. However, authors should openly acknowledge their informational advantage over WinCLIP in decision-making. While it's acceptable, clarity is vital. Additionally, I differ from the authors' portrayal of WinCLIP's prompt engineering. Notably, WinCLIP uses the same prompt template across categories, rather than tailoring individually. Also, the image-level results for WinCLIP in Table 2 should be 85.1%, but not 91.8%?

---

> > > ### Author Response · Authors · 2023-08-11
> > > **Thank you for your suggestions. We will incorporate them in the revised version.**
> > >
> > > Thank you for your timely reply and clarifications. We agree to incorporate your suggestions into the revised version of the paper. See details for each point.
> > >
> > > >  To Q1 and Q2
> > >
> > > We agree with your perspective on the difference between the object-level anomaly detection task and the pixel-level anomaly segmentation task, which helps make our paper and empirical study more precise. In our related work, especially in our few-shot and zero-shot AD paragraphs, we will highlight the difference between the two tasks and separate the discussion of related works per task. Specifically, we will summarize as follows. “Tasks in visual anomaly detection often involve object-level anomaly detection and pixel-level anomaly segmentation. The goal of the former is to differentiate image-level objects and the latter is to localize defects within an image. While meta-training for object-level anomaly detection is generally simpler (it is easy to find anomaly examples, i.e., other objects different from the normal one), meta-training for anomaly segmentation poses a harder task since image defects may differ from object to object (e.g., defects in transistors may not easily generalize to subtle defects in wood textures).” Thanks for the suggestions. We appreciate it!
> > >
> > > > To Q3
> > >
> > > A batch of patches refers to a collection of patches taken from a set of images that all share the same spatial position in the image. For example, we may stack the *top-left patch* of all testing `wood` images into a batch. Thank you for your careful review. We will add this information to the main paper and make it clear.
> > >
> > > > To Q4
> > >
> > > Thank you for your suggestions. We will discuss the different strategies for identifying “new normal distributions” across methods in more detail. Specifically, we will add the following sentences to our revised paper under the Zero-shot AD paragraph in the related work section: “While previous pre-trained CLIP-based zero-shot AD methods adapt to new tasks through informative prompts given by human experts, our method enriches the zero-shot AD toolbox with a new adaptation strategy without human involvement. Our approach allows the anomaly detector to infer the new task/distribution based on a mini-batch of samples.”
> > >
> > > To your second question, we double-checked the WinCLIP results in our paper and believe the reported results are correct. We borrowed WinCLIP image-level results from Table 1 in the WinCLIP paper, for which they refer to as “anomaly classification”. In Table 4 they report WinCLIP’s pixel-level anomaly segmentation results. We thank you again for the careful and helpful reviews.

---

> > > > ### Comment · Reviewer_hQFt · 2023-08-17
> > > >
> > > > Table 2 presents a scenario where the pixel-level results exhibit better performance than WinCLIP, while the image-level results lag far behind. It's worth considering the inadvertent swapping of ACR's image-level and pixel-level results. Interestingly, such a swap will make both the two results slightly beyond WinCLIP's performance. Could you double-check the correctness of ACR's results in Table 2? If there is no mistake, could you provide a possible explanation for the above concern? Or could you provide results on VisA dataset, aligned with WinCLIP? Will the results show a similar situation with MVTec?

---

> > > > > ### Author Response · Authors · 2023-08-19
> > > > > **We double-checked the correctness of ACR's results**
> > > > >
> > > > > Thank you for your attention. We doule-checked that our results are correct. Many other methods including SPADE (Cohen et al., 2020), Patch-SVDD (Yi et al., 2020), RegAD (4-shot, 8-shot, 16-shot, Huang et al., 2022), RotNet (Sohn et al., 2021), DisAug CLR (Sohn et al., 2021), GCPF (Wan et al., 2021), and PEDENet (Zhang et al., 2022), on the MVTec-AD dataset also share the same performance fashion with us: their pixel-level results are better than image-level results. The reasons of this phenomenon could be that 1) the extracted patch features are particulary suitable for pixel-level segmentations and are not perfectly good at image-level anomaly detection. Note that the image’s anomaly scores are derived from its pixels’ scores, i.e., taking the maximum of its pixel scores as the image’s score. In that sense, image-level anomaly detection performance on MVTec-AD is not directly optimized, but as a downstream task performed given the pixel-level anomaly scores; 2) In contrast, WinCLIP is based on CLIP, which is directly trained and tested on images as units rather than on image patches. This could explain why WinCLIP has better image-level performance but falls short on pixel-level segmentation.
> > > > >
> > > > > We would like to stress that the purpose of this MVTec-AD experiment is to demonstrate the broad applicability of our proposed method. Considering WinCLIP is the state-of-the-art method specifically designed for industrial anomaly detection, a direct application of our method on MVTec-AD leads to on-par performance with WinCLIP, with which we are satisfied. Our method is also competent at zero-shot AD on other data types like tabular data, for which WinCLIP cannot be applied. We thank again for the reviewer’s scrutiny.
> > > > >
> > > > >
> > > > > [Cohen, Niv, and Yedid Hoshen. "Sub-image anomaly detection with deep pyramid correspondences." arXiv preprint arXiv:2005.02357 (2020).]
> > > > >
> > > > > [Yi, Jihun, and Sungroh Yoon. "Patch svdd: Patch-level svdd for anomaly detection and segmentation." Proceedings of the Asian conference on computer vision. 2020.]
> > > > >
> > > > > [Huang, Chaoqin, et al. "Registration based few-shot anomaly detection." European Conference on Computer Vision. Cham: Springer Nature Switzerland, 2022.]
> > > > >
> > > > > [Sohn, Kihyuk, et al. "Learning and Evaluating Representations for Deep One-Class Classification." International Conference on Learning Representations. 2021.]
> > > > >
> > > > > [Wan, Qian, et al. "Industrial image anomaly localization based on Gaussian clustering of pretrained feature." IEEE Transactions on Industrial Electronics 69.6 (2021): 6182-6192.]
> > > > >
> > > > > [Zhang, Kaitai, Bin Wang, and C-C. Jay Kuo. "Pedenet: Image anomaly localization via patch embedding and density estimation." Pattern Recognition Letters 153 (2022): 144-150.]

---

> > > > > > ### Comment · Reviewer_hQFt · 2023-08-19
> > > > > >
> > > > > > Thank you for answering. This paper is overall a borderline paper. As the authors have addressed most of my concerns, I will slightly increase my score.

---

> > > > > > > ### Author Response · Authors · 2023-08-20
> > > > > > > **Thank you for raising the score**
> > > > > > >
> > > > > > > Thank you for engaging in the discussion and updating the score. We appreciate your suggestions.

---

### Official Review · Reviewer_CaFY · 2023-07-05

**Soundness:** 3 good
**Presentation:** 2 fair
**Contribution:** 2 fair
**Rating:** 5
**Confidence:** 5

**Summary:**

I think the paper has an interesting motivation but fails to deliver a convincing contribution. The method is incremental and lacks novelty and analysis.

**Strengths:**

The paper proposes a method for zero-shot batch-level anomaly detection using batch normalization and meta-training. The main idea is to train a deep anomaly detector on a set of interrelated data distributions and use batch normalization to adapt to unseen distributions at test time. The paper claims to achieve state-of-the-art results on image and tabular data. The paper is well-written and easy to follow

**Weaknesses:**

- The paper relies on three assumptions that may not hold in some real-world scenarios, such as the availability of a meta-training set of interrelated distributions, the batch-level anomaly detection setting, and the majority of normal data in each test batch. The paper does not justify the assumptions and design choices. The paper does not explain how realistic or generalizable these assumptions are, or how to relax or mitigate them in practice.

- The paper lacks novelty and theoretical analysis. The method is essentially a combination of existing techniques: batch normalization, meta-learning, and deep anomaly detection. The paper does not provide new insights or theoretical guarantees on why or how the proposed method works.

- The paper does not provide qualitative results to illustrate how its method works or what kind of anomalies it can detect. Such as some examples of test images or tabular data with their corresponding anomaly scores or segmentation masks generated by its method and compare them with those of the baselines. This would help to better understand the strengths and limitations of its method and provide some insights into its behavior and adaptability.

**Questions:**

Please see the weakness 3.

**Limitations:**

The paper lacks novelty and theoretical analysis. The method is essentially a combination of existing techniques: batch normalization, meta-learning, and deep anomaly detection. The paper does not provide new insights or theoretical guarantees on why or how the proposed method works.

---

> ### Author Rebuttal · Authors · 2023-08-09
>
> > The method is incremental and lacks novelty.
>
> We respectfully disagree. Before our work, zero-shot anomaly detection relied on large, pre-trained foundation models *specifically on image or language data*. We enrich the toolbox and provide a surprisingly simple solution. That is, one can use any existing deep anomaly detector (as long as their network structure contains batch normalization layers) plus proper training and evaluation schemes to get the zero-shot detection ability. In contrast to using foundation models, our approach may result in small, lightweight models that apply to data beyond text and natural images.  We discover the effect of batch normalization in zero-shot AD and propose to use meta-training to learn to use batch normalization to adapt to zero-shot AD. These points are also highlighted in our contribution list in Section 1 and the method explanation in Section 2.3.
>
> > lacks theoretical analysis
>
> Our paper does contain several theoretical results: we 1) derive generalization guarantees to unseen data distributions in Supplement A, 2) analyze why DSVDD fails when batch normalization is not used (Eq. 17), and 3) provide theoretical justifications for Assumption 3. They are mentioned in the main paper and we will refer to them more clearly.
>
> > Justification of the three assumptions.
>
> Thanks for pointing this out; we'll revisit our paper for improving clarity in this regard. Our Assumptions 2 and 3 are already justified in the paragraphs directly following their definition; Assumption 2 is the same assumption used in batch-level prediction work [10, 47, 51, 64, 73]. Assumption 1 is the same assumption typically chosen in the meta-learning literature [Finn et al., ICML 2017; Nichol et al., 2018]. In practice, the meta-training set can be generated using available covariates (e.g. for our tabular data experiment, we used the timestamps, in medical data, one could use data collected from different hospitals or different patients to obtain separate sets for meta-training, and in MVTec-AD, we used the other training classes except for the target class to form the training set.) We both empirically and theoretically justified Assumptions 3.
>
> > The paper does not provide qualitative results to illustrate how its method works or what kind of anomalies it can detect.
>
> Thanks for pointing this out–in fact, some of these qualitative results are already shown in the supplement. They are mentioned in the main paper and we will point this out more clearly. Figure 2 (supplement) shows why batch normalization is an adaptive zero-shot AD module and Figure 3 (supplement) visualizes how our method captures the normality (i.e., projects the normal samples towards the neighborhood of the “center” and pushes abnormal samples away) in practice on the Omniglot dataset.

---

> > ### Comment · Reviewer_CaFY · 2023-08-21
> >
> > Thanks to the authors for the rebuttal. For the response to novelty, the proposed method is a batch-level detector, while the pre-trained foundation models are more general. I think using meta-training to adapt BN for batch-level AD is incremental and wonder whether such kind of contribution reach the bar of NIPS. However, considering other reviewers' decisions, I will raise the score.

---

> > > ### Author Response · Authors · 2023-08-21
> > > **Could you update the score as promised in your last response**
> > >
> > > Thank you for reading our responses and agreeing to raise the score. We further address your new concerns as follows and hope to convince you of the novelty of our work. It will be a great help if you indeed increase the score.
> > >
> > > > The proposed method is a batch-level detector, while the pre-trained foundation models are more general
> > >
> > > We agree that batch-level prediction is one assumption our work makes, but pre-trained foundation models don't need. However, as we explained in our contribution list, our method enriches the zero-shot AD toolbox and has many benefits that pre-trained foundation models don't have. First, our method is applicable to many data types. For example, zero-shot AD in tabular data is one of our applications that foundation models cannot be applied. Second, our method doesn't involve humans at all but foundation models rely on carefully selected prompts provided by human experts to inform the model what the current task is. Third, the resulting model of our method is lightweight and takes little memory than large foundation models. In addition, experiments in our paper show that our lightweight method has on-par or even better performance than large pre-trained foundation models in specialized visual domains. In applications that are involved in these settings, our method is indeed more general than pre-trained foundation models.
> > >
> > > > I think using meta-training to adapt BN for batch-level AD is incremental and wonder whether such kind of contribution reach the bar of NIPS
> > >
> > > Applying meta-training to learn to use batch normalization layers to accomplish zero-shot AD is non-trivial. Previous to our work, meta-training and batch normalization were two individual concepts. Although one can directly apply batch normalization as a naive way to achieve zero-shot AD (illustrated by Eq. 4 in our paper), the performance is limited. Meta-training realizes the power of batch normalization layers in deep models. We theoretically proved the zero-shot AD generalization by applying meta-training with batch normalizations (Eq. 15 in supplement A), but also empirically demonstrated that meta-training improves the zero-shot AD performance by a large margin over only using batch normalization on pre-trained features (see baseline ResNet152). In addition, applying a meta-training dataset and batch normalization is independent of pre-trained foundation models, thus applicable to various data types and not restricted to images and texts as prescribed by foundation models.

---

### Official Review · Reviewer_Jqv8 · 2023-07-07

**Soundness:** 3 good
**Presentation:** 3 good
**Contribution:** 3 good
**Rating:** 5
**Confidence:** 4

**Summary:**

This paper introduces Adaptive Centered Representations (ACR) to tackle the zero-shot anomaly detection task. To this end, the ACR enables off-the-shelf deep anomaly detectors to generalize to unseen anomaly detection via training data distribution adaption concerning batch normalization and meta-learning. The numerical experiments of image and tabular data conducted on CIFAR100-C, OrganA, MNIST, Omniglot, MVTec AD, Anoshift, and Malware show the effectiveness of the proposed method.

**Strengths:**

•	This work introduces a simple yet effective training scheme, which integrates batch normalization and meta-learning, to enable zero-shot anomaly detection. The codes are included for reimplementation.

•	Experiments on several datasets show the superiority of the proposed method.


**Weaknesses:**

-	The proposed method relies on batch-level normalization. How the other normalizations, such as LayerNorm, InstanceNorm, and GroupNorm, affect the proposed training scheme is expected to be discussed.

-	The authors propose to apply batch normalizations in multiple layers for different anomaly scorers. Yet, the discussion and analysis of which layers and how their affections are not included.

-	The assumption 2 (A2) is about batch-level prediction, i.e., the anomaly scores are estimated based on batches. While carrying out prediction, is it able to automatically select the threshold for recognizing abnormal data? Or it needs to define a threshold per data class manually?


**Questions:**

The main concern of this paper is threshold selection while implementing anomaly detection. Besides, some discussions about normalizations and implementations are suggested.

**Limitations:**

The authors discussed their limitations in the manuscript.

---

> ### Author Rebuttal · Authors · 2023-08-09
>
> > Effects of other normalizations, such as LayerNorm, InstanceNorm, and GroupNorm
>
> We thank the reviewer for mentioning other normalization techniques. As follows, we report new experiments involving LayerNorm, InstanceNorm, and GroupNorm for zero-shot AD. We will add these discussions to the revised paper.
>
> We stress that, while these methods may have overall benefits in terms of enhancing performance, they do not work in isolation in our zero-shot AD setup. A crucial difference between these methods and batch normalization is that they treat each observation *individually*, rather than computing normalization statistics across a batch of observations. However, sharing information across the batch (and this way implicitly learning about distribution-level information) was crucial for our method to work.
>
> Our experiments (AUROC results in the table below) with DSVDD on the Omniglot dataset support this reasoning. Using these normalization layers in isolations yields to random outcomes (AUROC=50):
>
> |LayerNorm|InstanceNorm|GroupNorm|
> |---|---|---|
> |50.0$\pm$0.9|50.6$\pm$0.7|50.2$\pm$0.5|
> Rebuttal Table 1
>
> We also added a version of the experiment where we combined these methods with batch normalization in the final layer. The results dramatically improve in this case:
>
> |BatchNorm (BN) | LayerNorm + BN | InstanceNorm + BN | GroupNorm + BN|
> |---|---|---|---|
> |99.1$\pm$0.2|98.8$\pm$0.1|98.8$\pm$0.2|98.2$\pm$0.2|
> Rebuttal Table 2
>
> Experimental details:
> We use DSVDD as the anomaly detector and experiment on the Omniglot dataset. Each nonlinear layer of the feature extractor for DSVDD is followed by the respective normalization layer. We apply the same training protocol as Table 9 in the paper. For GroupNorm, we separate the channels into 2 groups wherever we apply group normalizations.
>
>
> > The authors propose to apply batch normalizations in multiple layers for different anomaly scorers. Yet, the discussion and analysis of which layers and how their affections are not included.
>
> We indeed work with different AD backbone models and generally apply batch normalization in all layers, which is standard and does not add much to the computational complexity. We agree that it is interesting to analyze whether batchnorm helps more in earlier or later layers. Our new experiments done above suggest that batchnorm in the final layer seems to work fairly well. We leave more systematic studies for the final version of the paper.
>
> > While carrying out prediction (for different data batches), is it able to automatically select the threshold for recognizing abnormal data? Or it needs to define a threshold per data class manually?
>
> As commonly done in the anomaly detection literature, we use AUROC as our evaluation metric [62, 60, 68, 49, 35] to assess the ranking between normal and abnormal data. That is, we use the anomaly score to rank the data and leave thresholding to the user. Threshold values depend on user requirements such as minimizing false positives, minimizing false negatives, adding sample rejection options, or setting the thresholds as quantiles. That said, an interesting avenue for future research is whether our meta-learning framework allows us to learn to predict suitable thresholds based on the meta-training set.

---

> > ### Comment · Reviewer_Jqv8 · 2023-08-14
> > **Official Comment by Reviewer Jqv8**
> >
> > Thanks to the authors for providing additional results. Since the additional results are conducted on the anomaly detection task on the Omniglot dataset, it seems that carrying out the batch-norm in the final layer is work. Does it also prefer the final layer while tackling the anomaly segmentation task on other datasets? It would be great if the relation between the layers using batch-norm and the various AD tasks could be further explored.

---

> > > ### Author Response · Authors · 2023-08-16
> > > **We conducted experiments to analyze the effects of the BatchNorm layer's position**
> > >
> > > We thank the reviewer for suggesting analyzing the effects of the BatchNorm (BN) layer’s position in various AD tasks. We conducted additional experiments on two visual anomaly detection tasks – anomaly segmentation on the MVTec-AD dataset and object-level AD on CIFAR100C. We used the same DSVDD model architectures as used in Tables 1 and 2 as the backbone model, except that we switched off BN in all but one layer. For anomaly segmentation, there are five possible BN layer positions; and there are four positions for the object-level AD model. We switched off the BN layers in all but one position and then re-trained and tested the model with the same protocol used in our main paper (For CIFAR100C, we tested the model with the test data anomaly ratio of 10%). We iterate this procedure across all available BN layer positions. We repeat every experiment with different random seeds five times and report the mean AUROC and standard deviation. The results are summarized in the tables below, where a smaller value of the BN position corresponds to earlier layers (close to the input), and a larger value corresponds to later layers close to the output. The final column is copied from our results in Tables 1 and 2 where BN layers are on all available positions. For both MVTec-AD and CIFAR100C, we average the performance across all test classes.
> > >
> > > Results on the two tasks have opposite trends regarding the effects of BN layer positions. Specifically, for anomaly segmentation on MVTec-AD, earlier BN layers are more effective, while for AD on CIFAR100C, later BN layers are more effective. This observation can be explained by the fact that anomaly segmentation is more sensitive to low-level features, while object-level AD is more sensitive to global feature representations. In addition, compared to the results in Tables 1 and 2 (copied to the last column in the table below), our results suggest that using BN layers at multiple positions does help re-calibrate the data batches of different distributions from low-level features (early layers) to high-level features (late layers) and shows performance improvement over a single BN layer.
> > >
> > >
> > > *MVTec-AD*:
> > > |BN position|1|2|3|4|5|(1,2,3,4,5)|
> > > |---|---|---|---|---|---|---|
> > > |Pixel-level|80.8$\pm$1.9|69.6$\pm$1.4|73.9$\pm$0.9|63.6$\pm$1.6|60.9$\pm$0.8|92.5$\pm$0.2|
> > > |Image-level|74.7$\pm$0.9|59.2$\pm$1.6|63.6$\pm$1.3|65.5$\pm$1.2|65.4$\pm$1.3|85.8$\pm$0.6|
> > >
> > > *CIFAR100C*:
> > > |BN position|1|2|3|4|(1,2,3,4)|
> > > |---|---|---|---|---|---|
> > > |AUROC|61.4$\pm$0.5|61.0$\pm$0.9|68.2$\pm$0.9|68.9$\pm$1.1|85.9$\pm$0.4|

---

### Author Rebuttal · Authors · 2023-08-09

We thank the reviewers for all the valuable suggestions on additional ablations. Based on their inputs, we have run additional experiments to study different normalization layers (Rebuttal Tables 1 and 2), studying different values for the hyperparameter $\pi$ (Rebuttal Table 3), meta-OOD performance (Rebuttal Table 4), additional varying number of training classes (Rebuttal Table 5), N-vs-rest setting (Rebuttal Table 6), and tiny batch sizes (Rebuttal Table 7). The most interesting findings will be included in the main paper to complement the extensive empirical study: we have studied three different tasks (image-level anomaly detection, anomaly segmentation, and anomaly detection in tabular data)  where each task contains multiple datasets and we compared at least six different methods for each task.

---

### Decision · Program_Chairs · 2023-09-21

**Decision:**

Accept (poster)

**Comment:**

This paper tackles a relatively unexplored meta-learning setting for anomaly detection. The reviewers appreciated the simple approach and the strong performance. Furthermore, the AD community has recently been exploring new modes of supervision for anomaly detection given the impossibility of fully-unsupervised AD. Meta-learning is a promising direction and the simplicity of this approach may encourage further work in this direction. The rebuttal convinced all reviewers to update their reviews to accept or borderline-accept. The AC concurs and recommends acceptance as well.